# Transcriptome-wide gene-gene interaction associations elucidate pathways and functional enrichment of complex traits

**Luke M. Evans**[1,2]*, **Christopher H. Arehart**[1,2], **Andrew D. Grotzinger**[1,3], **Travis J. Mize**[1,2], **Maizy S. Brasher**[1,2], **Jerry A. Stitzel**[1,4], **Marissa A. Ehringer**[1,4], **Charles A. Hoeffer**[1,4]

**1** Institute for Behavioral Genetics, University of Colorado Boulder, Boulder, Colorado, United States of America, **2** Department of Ecology & Evolutionary Biology, University of Colorado Boulder, Boulder, Colorado, United States of America, **3** Department of Psychology & Neuroscience, University of Colorado Boulder, Boulder, Colorado, United States of America, **4** Department of Integrative Physiology, University of Colorado Boulder, Boulder, Colorado, United States of America

\* luke.m.evans@colorado.edu

**Data Availability Statement:** All data used are from publicly available repositories, accessible publicly or with appropriate approval from the

## Abstract

It remains unknown to what extent gene-gene interactions contribute to complex traits. Here, we introduce a new approach using predicted gene expression to perform exhaustive transcriptome-wide interaction studies (TWISs) for multiple traits across all pairs of genes expressed in several tissue types. Using imputed transcriptomes, we simultaneously reduce the computational challenge and improve interpretability and statistical power. We discover (in the UK Biobank) and replicate (in independent cohorts) several interaction associations, and find several hub genes with numerous interactions. We also demonstrate that TWIS can identify novel associated genes because genes with many or strong interactions have smaller single-locus model effect sizes. Finally, we develop a method to test gene set enrichment of TWIS associations (E-TWIS), finding numerous pathways and networks enriched in interaction associations. Epistasis is may be widespread, and our procedure represents a tractable framework for beginning to explore gene interactions and identify novel genomic targets.

## Author summary

We developed a new method to comprehensively test associations of all pairwise gene-gene interactions with complex traits using imputed expression. We applied the method to 12 complex traits in humans across four tissues or cross-tissue expression measures. We found widespread evidence that gene-gene interactions influence traits, and that accounting for interactions identifies loci not previously identified in traditional single-locus association tests, because the interactions mask the main effects when tested in isolation. We next introduced a gene set analysis to test enrichment of interaction associations in pathways and cell types and identify several gene sets within which gene interactions are enriched in the associations with complex traits. Our analyses identify core hub genes that appear to integrate signals across multiple pathways, providing new biological insight into the genetic influences on these traits. Our findings also confirm the role of gene interactions in complex traits, which has long been hypothesized but never before

repositories: MsigDB: https://www.gsea-msigdb. org/gsea/msigdb/; FUSION: http://gusevlab.org/ projects/fusion/; UKBiobank: https://www. ukbiobank.ac.uk/; dbGaP: https://www.ncbi.nlm. nih.gov/gap/, including ARIC (phs000280.v7.p1), GERA (phs000674.v3.p3, phs000788), and NESARC-III (phs001590.v2.p1); Genes for Good: https://genesforgood.sph.umich.edu/. Meta-analyzed TWIS summary statistics and SNPxSNP interaction summary statistics are available on Dryad (https://doi.org/10.5061/dryad.866t1g1tw), and scripts to perform TWIS and E-TWIS can be found on GitHub (https://github.com/evanslm/ TWIS), and TWAS at https://github.com/gusevlab/ fusion_twas.

**Funding:** LME was supported by the University of Colorado Boulder Institute for Behavioral Genetics the National Institutes of Health AG046938-06, DA044283-01, and MH100141-06; TJM was supported by DA017637; MAE was supported by DA051937 and AA026733, and CAH was supported by AG064465 and the Linda Crnic Institute for Down Syndrome. The funders had no role in study design, data collection and analysis, decision to publish, or preparation of the manuscript.

**Competing interests:** The authors have declared that no competing interests exist.

comprehensively tested due to the computational burden required, but which our new approach can efficiently and effectively deal with.

## Introduction

Genome-wide association studies (GWASs) have identified numerous individual loci that affect complex traits [1,2]. Recent developments in transcriptome imputation and transcriptome-wide association studies (TWASs) have enhanced our understanding of complex traits by providing biologically plausible mechanisms of action for associated genes and improving power by aggregating small individual variant effects on gene expression to identify associations [3–5]. The overwhelming majority of these identified loci have been detected using an additive model of alleles at individual loci [1,2,6].

While GWAS and TWAS have expanded our understanding of the genetic architecture underlying complex traits, a fundamental, unresolved question is to what extent non-additive effects contribute. Specifically, epistasis, defined as the statistical dependence of the allelic effects at one locus on the genotype at another locus [7], may influence quantitative traits [7–10]. It is increasingly clear that complex traits are exceedingly polygenic, with influences from many complex regulatory and molecular pathways, and even chromosomal three-dimensional structure [11–13]. Such complexity makes gene interactions likely to exist and these interactions have been demonstrated using several model systems and organisms [7–9,14,15]. While there has been debate over whether non-additive genetic variance is a major contributor to heritability [6,16–20], non-additive gene action contributes to additive as well as non-additive variance components [21,22], and thus epistatic gene action could still play a role in the underlying genetic architecture of complex traits, even for traits of largely additive genetic variance. Identifying gene-gene interactions and the pathways and networks in which they occur will provide a critical context for understanding the biology of complex traits [7,10]. Ascertaining the prevalence and magnitude of epistasis would also clarify interpretation of family-based, and specifically twin-based, estimates of heritability, which may be inflated by non-additive variance in combination with maternal or environmental effects [16,20].

Despite the likely importance of epistasis, genome-wide interaction tests remain rare. Computational burden, correlation among predictors (leading to false positive epistatic associations [23,24]), and interpretability are key challenges to genome-wide, exhaustive tests of epistasis [7,25–27]. Perhaps the greatest challenge is that the sheer number of variants available in imputation panels (10M+) leads to tens of billions of pairwise tests, which despite recent methodological advances [28,29] remains prohibitive. Many address this through two-stage approaches, in which the predictors are filtered in some way prior to testing epistasis among the retained predictors [26,30]. Often, interactions are only tested between loci that are significant in single-locus GWAS or phenotypic variance test effects or are based on hypothesized pathways or networks. While such methods improve feasibility by reducing the number of tests, they constrain the ability to detect novel epistatic effects or new pathways and networks involved in complex traits [8], and in some cases, do not indicate whether the interactor effect is an environment or a second gene [30,31]. Similarly, if a strong interaction between two loci exists, the main effects estimated in a single-variant GWAS could be muted [7], reducing the likelihood of identifying such interactions in two-stage approaches. Thus, exhaustive approaches are preferable to two-stage or filtered approaches.

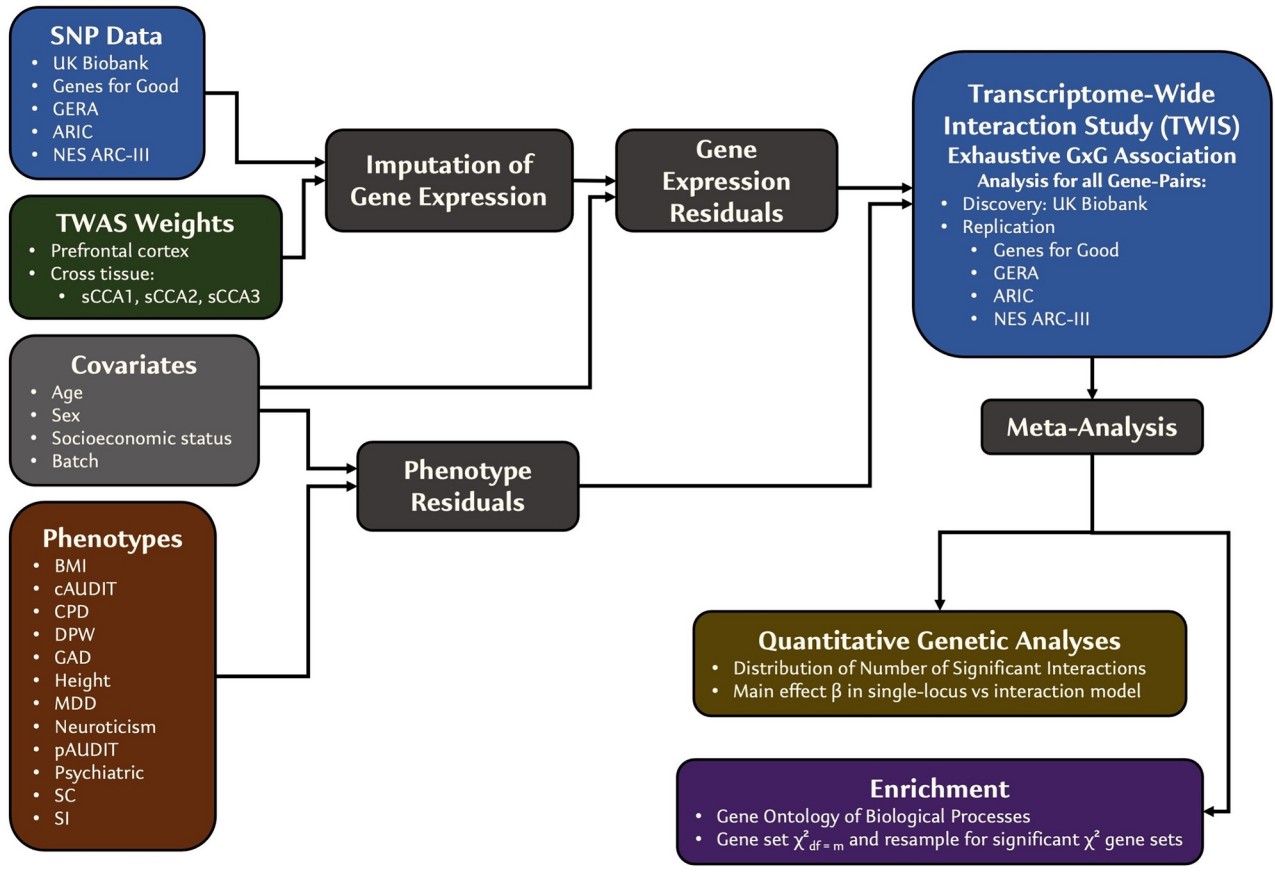

**Fig 1. Overview of TWIS approach.**

Here, we report an innovative approach using imputed transcriptomes to perform exhaustive transcriptome-wide interaction studies (TWISs) for multiple traits across all pairs of genes expressed in several tissue types (Fig 1). Using imputed transcriptomes, we provide an approach to simultaneously reduce the computational challenge and improve interpretability, while also aggregating small interaction effects of individual variants via gene expression to improve statistical power to detect interaction associations. We begin by performing extensive simulations to validate the TWIS approach and develop standardized analytic procedures, including power analyses, multiple test correction thresholds, and pruning on LD that can lead to false positives. Importantly, we find that unmodeled interactions can also produce false negatives for main effects such that TWIS both identifies epistatic effects and identifies previously unassociated loci. Finally, we develop and validate Enrichment TWIS (E-TWIS), a novel method for aggregating genome-wide gene-gene interactions with respect to *a priori*-defined gene sets to understand the specific functional networks enriched for epistatic effects. In an empirical application, we identify several replicated, significant interactions and numerous functional gene sets and brain cell types that are enriched in interaction associations. Epistasis is likely a major source of phenotypic variation in complex traits, and the analytic procedures and results presented here reflect a computationally and statistically tractable framework for beginning to unpack these interactive effects.

## Results

### TWIS Approach—Simulation and validation

Fig 1 is a diagram of our overall transcriptome-wide interaction study (TWIS) approach. We leveraged a total of five cohorts to perform discovery and replication TWIS of 12 complex traits, including biometric, substance use, and psychiatric traits (defined in S1 Table and S1 Note). We used the UK Biobank as the discovery cohort to identify significant interactions (N = 53,880–329,705) and used the remaining 2–3 cohorts (depending on the trait) as an independent replication sample (N = 8,718–61,531). Following standard quality control (see Methods and S1 Note), we imputed gene expression in each cohort for the prefrontal cortex (PFC, $m$ = 14,729 genes) using FUSION [3,4]-generated TWAS weights from the PsychENCODE consortium [32]. The PFC was chosen because of the importance of neurocognitive functions in many of the traits we examined (e.g., psychiatric and substance use traits) and because it is currently the largest available brain reference panel with expression TWAS weights. Because of the large number of possible tissues relevant to complex traits, we also used cross-tissue expression weights from the first three sparse canonical correlation axes (sCCA1-3) of Feng et al.[33] ($m$ = 13,242; 12,521; and 12,032). Here, we include tests using all tissues in all traits for completeness, but a reasonable approach to reduce the overall number of tests would be to perform TWIS using only expression in biologically relevant tissues, cross-tissue expression measures, or in those tissues with, for example, significant LDSC $h^2_{SNP}$ enrichment [34,35] for the trait of interest.

Correctly accounting for covariates and possible confounding effects in interaction associations requires including all covariate-by-main effect interactions [36], which quickly increases computational time with numerous covariates and categorical factor levels. Therefore, following QC and expression imputation, we residualized phenotype and imputed expression on covariates prior to performing the gene-gene interaction associations (see Methods). This residualization does not affect the false positive rate of the interaction test relative to a full model (S2 Table). The cohorts differed in the specific measures available, but included measures of age, sex, educational attainment, income or socioeconomic status, genotyping batch (where available), and the first 10 genomic principal components. When performing 10s of millions of tests, this residualization step substantially decreased the total computation time while estimating unbiased gene-gene interaction effects. Following this step, we used a parallelization procedure to divide all $\binom{m}{2}$ pairwise interactions across multiple compute nodes for each trait and each tissue, testing the simplified model,

$$y_{resid} = \mu + \beta_1 T_{1resid} + \beta_2 T_{2resid} + \beta_{int} T_{1resid} * T_{2resid} + \varepsilon \qquad (1)$$

where $y_{resid}$ is the phenotype residualized on the covariates; $T_{1resid}$ and $T_{2resid}$ are the imputed expression of genes 1 and 2, respectively, residualized on the covariates; $\mu$ is the intercept; $\beta_1$ and $\beta_2$ are the main effects of $T_{1resid}$ and $T_{2resid}$; $\beta_{int}$ is the gene expression interaction effect on the phenotype; and $\varepsilon \sim N(0,\sigma^2)$ is the error. We emphasize that this model does not require physical interaction of gene products, only that the association of expression of one gene is affected by that of another. Such interactions could include physical interaction, but also other mechanisms, such as stoichiometric relationships within molecular pathways.

### Power and Significance Thresholds

We performed a series of simulations to estimate power to detect interactions in the context of imperfect expression imputation (where imputation $r^2<$expression heritability, the maximum accuracy of the genetic prediction) across a range of epistasis effect sizes, define the

appropriate $\alpha$ for genome-wide multiple test correction in the context of many millions of individual tests, and assess the role of LD in influencing interaction tests (see Methods and S1–S8 Figs). Consistent with prior findings [23,24], we find that pairs of genes with imputed expression correlations ($|r| > 0.1$) or those physically nearby produce inflated type I error for identifying interaction effects. True, nearby interacting loci do exist, such as *HLA* region variant interactions influencing multiple sclerosis [37,38], and linked interacting loci have been hypothesized as a source of genetic variance [39,40]. We note that gene pair correlated expression may result from LD between causal eQTLs for each gene, as well as shared eQTLs affecting both genes directly [41]. Given the drastic increase in false positive rates due to correlated predictors, we view excluding these nearby or genes with correlated expression as a reasonable tradeoff.

Within each phenotype, we applied a significance threshold of $p<5.86e\text{-}10$ (see Methods) while also excluding from further analysis any pairs of genes whose imputed expression $|r|>$ 0.05 (more conservative than the $|r|>0.1$ suggested by simulations) at the discovery stage (UK Biobank sample) or those within 1MB of each other. In independent replication, we applied, first, this correction within each phenotype and tissue to interactions identified within the discovery cohort, and second, a nominal $p<0.05$ as suggestive evidence of replication. Finally, we meta-analyzed [42] all cohorts together (discovery + replication) for use in functional and pathway enrichment analyses. See S3 Table for a list of all thresholds applied and notes about their context.

## TWIS Associations—Empirical Results

We applied TWIS to 12 traits (height, BMI, cigarette smoking initiation [SI], smoking cessation [SC], heavy vs. light cigarettes per day [CPD], major depressive disorder [MDD], generalized anxiety disorder [GAD], neuroticism, cross-trait psychiatric disorders [PSYCH], problematic alcohol use [pAUDIT], alcohol consumption [cAUDIT], and drinks per week [DPW]; see S1 Note for full phenotype and cohort descriptions). Across all traits and tissues, 16 pairwise interactions were significant ($p<5.86e\text{-}10$) at the discovery stage, only one of which replicated ($p<0.05$) in independent replication datasets in the same direction. Of these 16, four remained significant ($p<5.86e\text{-}10$) in the final (discovery + replication) meta-analysis (Table 1 and Figs 2 and S9–S20). One additional interaction was significant when all cohorts were meta-analyzed, but not in discovery or replication. S21–S25 Figs for figures of the raw phenotype plotted against imputed expression of both genes, and S4 Table for all pairs that were significant at any stage.

We subsequently tested additive-by-additive SNPxSNP interactions, using a similar residualization approach, of all pairs of SNPs within 500KB of each gene, for the five gene pairs in Table 1 that were significant in the final meta-analysis. No interaction, in individual cohorts or meta-analyzed across all cohorts for each trait, reached our multiple testing threshold of $p<5.86e\text{-}10$, but several pairs approached this ($p<5e\text{-}8$), suggesting that with sufficient sample size and power, TWIS is a reasonable approach to identify a restricted set of genes around which all SNPxSNP interactions can be tested, perhaps including multiple forms of interaction (additive x additive, dominance x dominance, etc.), likely in part by aggregating individual additive expression effects of SNPs together.

Of the four interactions significantly associated with pAUDIT in the final meta-analysis (Table 1), three involved *PRKCG* imputed prefrontal cortex expression, interacting with *WNT6*, *MAP7*, and *SEZ6L2*. Higher levels of imputed *PRKCG* expression were associated with stronger (more positive) effects of the interacting gene (S22–S24 Figs), consistent with the positive interaction term. Notably, WNT is known to modulate PKC localization and activity via

**Table 1. Interaction associations of pairs that reached $p \leq 5.86e\text{-}10$ in the final combined meta-analysis.** Replication and final combined results indicate the meta-analyzed $Z$ scores. See S4 Table for all pairs that were significant at any stage.

| Trait & Expression Tissue<br>Gene Name, ENSGID, chromosome and midpoint bp location | | | Discovery | | | Replication | | Final Combined | | |
|---|---|---|---|---|---|---|---|---|---|---|
| Gene 1 | Gene 2 | Expression $\rho$ | $\beta$ | SE | $p$ | Z | $p$ | Z | $p$ | Direction |
| *pAUDIT, Prefrontal Cortex Expression* | | | | | | | | | | |
| *PRKCG* (ENSG00000126583, 19:54296675) | *WNT6* (ENSG00000115596, 2:219731750) | 0.000 | 0.068 | 0.011 | 2.86E-10 | 1.884 | 0.060 | 6.483 | 9.01E-11 | +++ |
| *PRKCG* (ENSG00000126583, 19:54296675) | *MAP7* (ENSG00000135525, 6:136767916) | -8.39E-05 | 0.040 | 0.006 | 1.51E-10 | 2.432 | 0.015 | 6.816 | 9.36E-12 | +++ |
| *CENPN* (ENSG00000166451, 2:122008473) | *TFCP2L1* (ENSG00000115112, 16:81053411) | 0.003 | -0.157 | 0.022 | 4.14E-13 | -1.535 | 0.125 | -7.171 | 7.43E-13 | -+- |
| *PRKCG* (ENSG00000126583, 19:54296675) | *SEZ6L2* (ENSG00000174938, 16:29896674) | -0.002 | 0.105 | 0.017 | 1.89E-10 | 1.82 | 0.069 | 6.511 | 7.45E-11 | +++ |
| *GAD, Cross-Tissue Expression, sCCA3* | | | | | | | | | | |
| *MTMR10* (ENSG00000166912, 15: 30965284) | *SEPHS1* (ENSG00000086475, 10: 13332863) | 0.0004 | 0.0123 | 0.002 | 5.79E-09 | 6.359 | 2.03E-10 | 6.359 | 2.03E-10 | +++ |

G-protein- and Ca2+-dependent mechanisms [43,44]. MAP7 is known to directly interact with PKC signaling [45] and has a role in axon collateral branching [46,47]. SEZ6L2 is a cell-surface protein that regulates neurogenesis and differentiation through adducin signal transduction [48], which is a substrate for PKC.[49] The fourth interaction associated with pAUDIT was *TFCP2L1xCENPN*, which was found to have a negative interaction term, consistent with a reduced effect (less positive slope) of *CENPN* expression at higher levels of imputed *TFCP2L1* expression (Table 1 and S25 Fig). *TFCP2L1*, which is down regulated in cells exposed to alcohol [50], regulates transcription involved in pluripotency and cell renewal and is also involved in the WNT pathway [51]. CENPN is a histone that forms a complex with other histones in the presence of DNA and locates at the centromere, forming kinetochores [52]; their interaction may reflect effects on neurogenesis or neural cell types from a brain stem cell.

*MTMR10xSEPHS1* was significantly associated with GAD in the final meta-analysis (Table 1 and S21 Fig), in which a stronger effect of imputed *MTMR10* was associated with higher *SEPHS1*. Both expressed in glial cells, *MTMR10* is in a locus associated with schizophrenia and dendritic growth deficiency [53,54], substance use disorders and related behavioral traits [55], while *SEPHS1* deregulation has been reported in rats under chronic stress [56]. SEPHS1 influences selenium metabolism pathways, deficiencies in which lead to oxidative stress [57] and increased inflammation and degradation of extracellular matrix [58]. MTMR10 plays a role in the extracellular matrix, including in neurons and protects dendrites in response to oxidative stress [59]; their interaction may relate to regulation of inflammation and stress response.

The limited number of significant interaction associations was not surprising given the low power to detect small effect sizes, particularly when expression imputation is imperfect and with stringent multiple test correction (S1–S4 Figs). As in single-locus GWAS, we anticipate additional, replicated loci to be identified with larger GWAS and expression reference panels, because imperfect expression imputation sharply reduces power.

For genes involved in at least one suggestive ($p \leq$1e-5) interaction association, we found that across all traits, the number of interactions per gene followed a power-law distribution, with the majority of genes participating in only one or two interactions, but a few involved in many (S26–S37 Figs and S5 Table). These "hub" genes (examples in Fig 3) are highly connected genes that represent logical targets for functional follow-up and characterization as hubs of interactions with many genes, integrating signals throughout pathways. While they may be poor drug targets as critical bottlenecks that impact multiple traits, identifying the

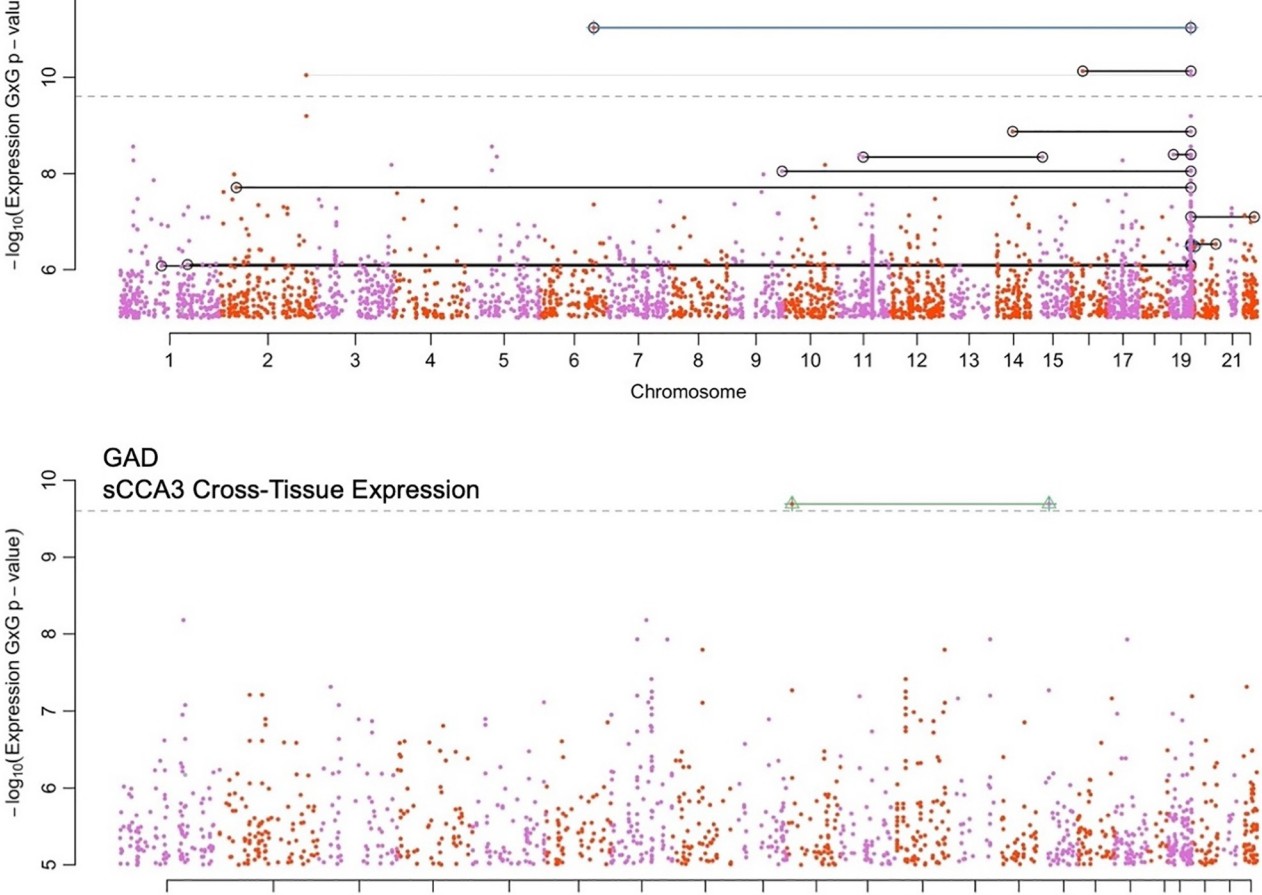

**Fig 2. Boulder plot of pAUDIT (top) and GAD (bottom) interaction association p-values using imputed transcription.** Shown are the results from the final meta-analysis of all data. In these plots, each interaction test is indicated by two points, located at their physical chromosomal positions. Pairs with significant interactions are connected by lines. Peaks, such as the peak on Chromosome 19 in the top figure, indicate strong interactions with many other genes, i.e., a hub gene (see Fig 3 as well). Black lines connect pairs that surpassed p<5.86e-10 in the discovery cohort (UKB), green and blue lines connect pairs of loci with FDR q<0.05 or nominally significant interaction (p<0.05) in the replication cohort, and gray lines connect pairs of genes with p<2.5e-10 in the final meta-analysis. For clarity, only interaction associations with p<1e-5 are shown. Numerical results of genes reaching significance are presented in Table 1.

genes they interact with could be a useful approach to find specific targets to modulate in developing therapeutics. The gene with the most interactions was, with pAUDIT using PFC expression, *FOLH1B*, an untranslated pseudogene previously associated with psychiatric disorders [60] and BMI [61]. *PRKCG*, noted above, was the second most interacting gene, again with pAUDIT using PFC expression. The glutamate receptor *GRIK1* had the most interactions associated with CPD but was not identified in single-locus GWAS by the GSCAN study [62], despite GSCAN's much larger sample size and higher statistical power, demonstrating that novel associated genes can be found using TWIS, and the possible role of glutamate and excitatory neurotransmitters in smoking [63]. *RRAGA*, which regulates [64] the mTOR signaling cascade [65] that may have a role in the antidepressant effects of NMDA antagonists [66], was

## All pairs TWIS Interactions, *p*<1e-6

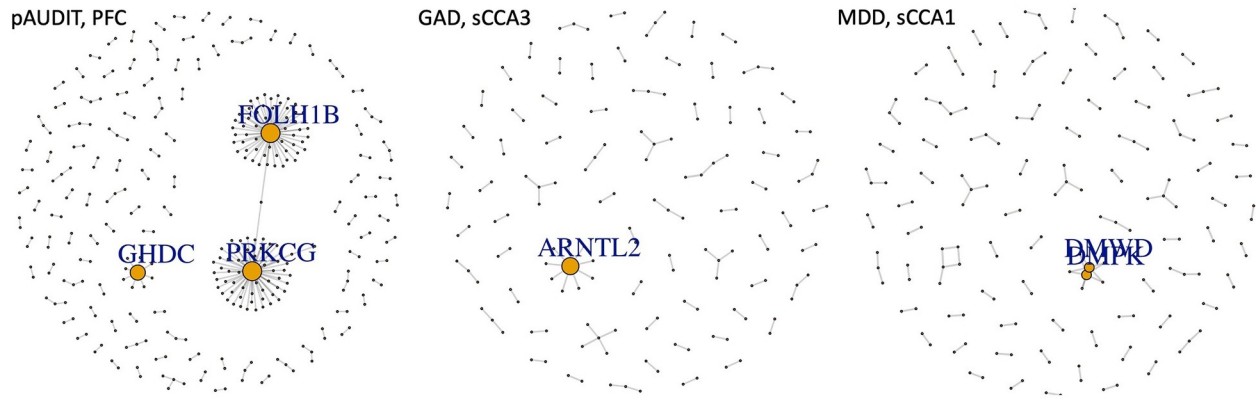

## Gene set TWIS Interactions, PFC expression, *p*<1e-3

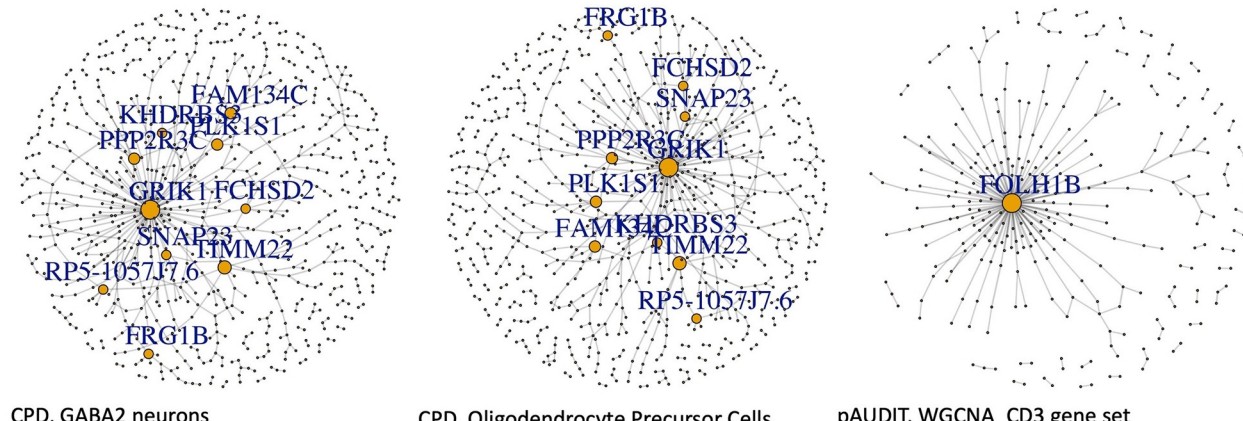

**Fig 3. Networks of TWIS associations for selected traits and gene expression in specific tissues, either based on all pairs with *p*<1e-6 from the exhaustive, genome-wide TWIS (top), or within specific gene sets applying a nominal *p*<1e-3 threshold (bottom).** P-value thresholds were chosen to best visualize clusters. Genes with degree≥5 are labeled, and size of points is proportional to node degree.

the most interacting gene associated with GAD, highlighting the possible role of the mTOR pathway for internalizing disorder treatment. From a genetic architecture perspective, these findings support a long-standing hypothesis that while epistasis is common, most genes will interact with a limited number of other genes [7]. They also support an omnigenic model [67] of architecture, where core or hub genes interact with and incorporate the regulatory effects of many peripheral genes. TWIS may identify such core or hub genes more directly than single gene association models.

Given our exhaustive, all-pairs TWIS for multiple traits, we were also able to test whether genes with evidence of interaction association would have been identified in a single gene TWAS, as it is hypothesized the effect sizes of a locus could be diminished when analyzed individually if the gene's effect depends on an interaction with another [7]. For example, *GRIK1*, noted above as the gene with the most interactions associated with CPD, would not have been identified in a single locus TWAS using the same dataset ($p = 0.35$), nor was it identified in the largest CPD GWAS to date [62]. Using pairs of suggestive ($p$<1e-5) interaction associations in the combined meta-analysis, we estimated that, on average, only 3% (SD = 6.8%) of the unique

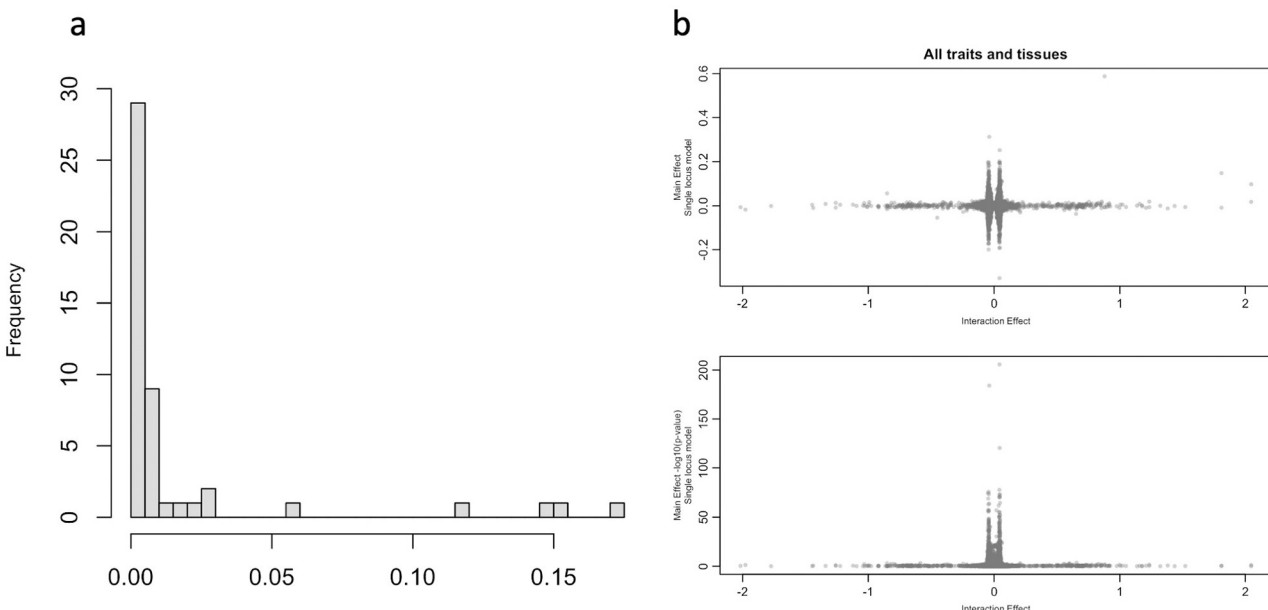

**Fig 4.** (a) Proportion of genes identified within suggestive interaction associations (p≤1e-5) that would have been identified using the same threshold in a single gene TWAS. Data in S3 Table. (b) Relationships of TWIS interaction effect sizes and main effect sizes of the same genes from TWAS (single locus model). Estimates of effects from all genes identified in TWIS included across traits and tissues, but each TWIS-identified gene is included only once per trait and tissue combination, even if a gene interacted with multiple other genes.

genes identified in TWIS would have been identified using a single gene TWAS (Fig 4a and S6 Table). As an example, of the 1106 unique genes in 655 pairs identified with GAD TWIS associations using PFC expression, none would have been associated in single locus models. Similarly, of the 981 unique genes in 547 interacting pairs associated with BMI using PFC expression, only 25 would have been identified in single-gene TWAS (S6 Table). This results from reduced effect sizes in the single gene TWAS for genes with the largest interaction effects (Fig 4b). This is consistent with the hypothesis that when a gene interacts with many others, its estimated effect in a single locus model may not be strong [7], and it highlights the fact that novel loci may be identified using an exhaustive, all-pairs TWIS relative to single-locus TWAS or GWAS, with *GRIK1*, noted above, an example.

### Functional and pathway interaction enrichment

We developed Enrichment-TWIS (E-TWIS) to assess the strength of interaction associations among genes within *a priori* defined gene sets of interest, rather than individual pairs of genes, including multiple functional pathways and networks. We first used a measure similar to network connectivity [68] to use $\chi^2$ tests to efficiently test enrichment of approximately 8,000 gene sets. This was anti-conservative for large (n>150 genes) gene sets, where we used a random resampling approach to confirm enrichment (S38–S40 Figs). The resampling represents a competitive test (*sensu* [69]) of enrichment relative to background epistatic interactions, and in practice produced qualitatively similar results. We advocate an approach of efficiently testing many gene sets via $\chi^2$ tests and using resampling to confirm significant gene set enrichment or to test sets of particular interest.

Gene sets we tested (~8,000) included the weighted gene coexpression network analysis (WGCNA) modules in PFC expression data [32,70], many sets defined in the Molecular

Signatures Database (MsigDB) [71], and genes specifically expressed within individual cell types within multiple brain regions and subsets intolerant to protein-truncating mutations [72]. These represent a wide range of types of gene sets, across a wide variety of functional pathways, tissue expression specificity, and possible interactions (e.g., WGCNA modules), for an exploratory analysis of interaction enrichment.

We identified 50 significantly associated (FDR<5%) gene sets across all traits and expression tissues (Figs 3 and 5 and S7–S8 Tables). Among the associated gene sets, a common theme for several traits, notably GAD, PSYCH, neuroticism, CPD, and alcohol use, was enrichment of sets related to immune system and inflammation pathways. For neuroticism, we identified *STAT1* transcription factor binding sites as enriched, which regulates cellular responses to interferons, cytokines, and other growth factors, and plays a role in immune response. Genes involved in immune system function (upregulated in T cells relative to B cells) were enriched in GAD, together suggesting the importance of immune system and inflammatory pathways for anxiety-related traits. Genes with expression influenced by *FOXP3*, which regulates immune system response including *IL2*, were enriched in psychiatric case epistatic interactions.

Evidence of cell signaling pathway enrichment was also found, such as glutamate receptor genes for GAD (S8 Table). G-protein mediated event genes were enriched for pAUDIT, which includes signal transduction at the synapse, and is consistent with the *WNT6-PRKCG* interaction noted above (and possible immune function). Gene interactions within the deubiquitination REACTOME pathway were associated with pAUDIT, suggesting the importance of post-

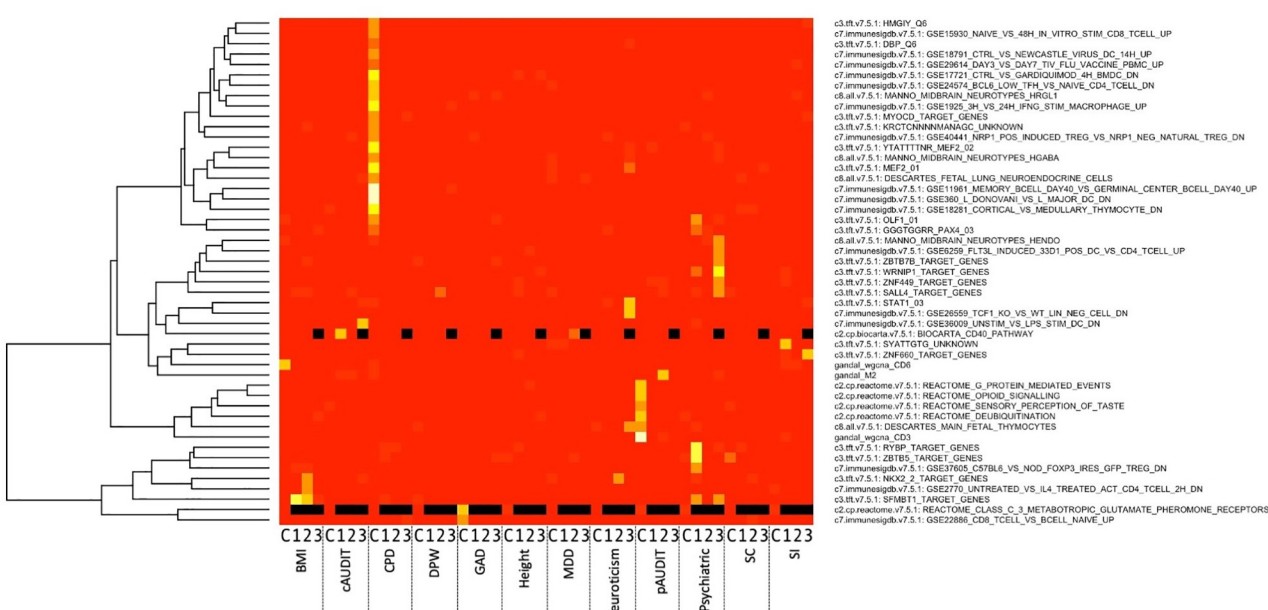

**Fig 5. Gene set enrichment across all tissues and traits for those sets with at least one significant test (FDR<5%).** Black indicates that the gene set association was not evaluated for that tissue and trait combination. X-axis shows the trait and tissue, where C indicates PFC and 1–3 represent the cross-tissue sparse canonical correlation axes 1–3. Phenotype details are in the S1 Note and S1 Table.

translational modification in alcohol use as has been hypothesized [73], and highlighting the need for additional 'omics integration into such analyses. Notably, three of the coexpression network modules identified by Gandal et al.[32,70] were associated with BMI or pAUDIT. The gene M2 network (associated with pAUDIT) was found to be downregulated in oligodendro-cytes in bipolar and schizophrenia cases [32], while the CD3 module (also associated with pAUDIT) was found to be enriched in oligodendrocytes [70], suggesting a role for glia.

Among gene sets specifically expressed in individual cell types [72], we found enrichment of many traits for interactions in both excitatory and inhibitory neurons, with a number of GABAergic neuron enrichments (Figs 3 and 6 and S9–S10 Tables). Notably, excitatory neu-rons were strongly enriched in CPD, supporting the individual strong interactions of *GRIK1* noted above. Oligodendrocytes and/or their precursor cells were enriched in BMI, CPD, height, MDD, and pAUDIT, highlighting a role of non-neuronal cells in several traits.

## Discussion

Here, we present the first, to our knowledge, fully exhaustive transcriptome-wide interaction study of all pairwise gene interaction associations. We confirmed several long-standing expec-tations of quantitative genetics, including that most genes have only a few interactions while a few 'hub' genes contain many, and that for genes with strong gene-gene interactions, estimated effects from a single-locus models are weaker. These two findings imply that epistasis may be frequent, and key hub genes may yet be identified. These results also suggest that exhaustive

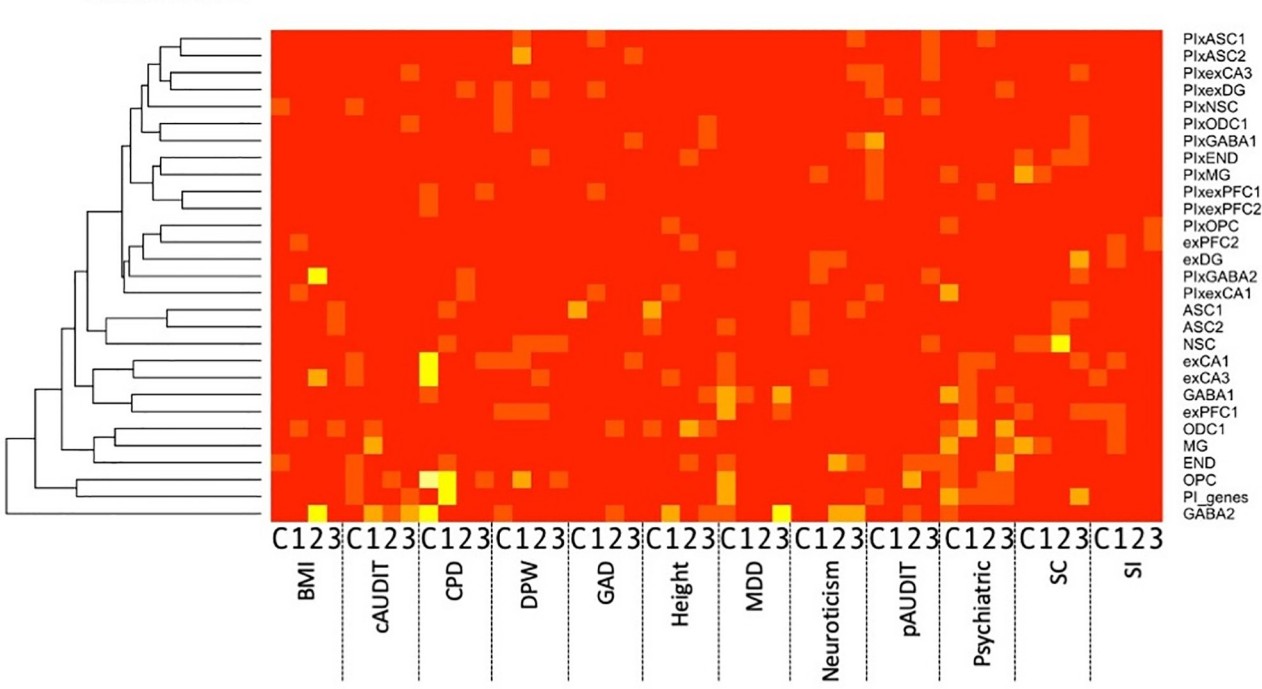

**Fig 6. Neuronal cell type [72] gene set interaction association enrichment across all tissues and traits.** X-axis shows the trait and tissue, where C indicates PFC and 1–3 represent the cross-tissue sparse canonical correlation axes 1–3.

interaction studies are needed rather than two-stage or variance models, which are efficient but may fail to detect real interactions. TWIS is an efficient way to both reduce the overall number of tests (on the order of 1e8 rather than 1e12 SNPxSNP tests) and improve power by integrating small individual SNP effects on expression. Although other approaches have been proposed [30,74,75], we have built upon previous findings suggesting epistasis is important for complex traits and provide a novel framework in which to exhaustively search all pairwise gene-gene interactions.

We also present findings of power analyses and type-I error, which verify both low power, as expected in interaction tests, as well as a need for stringent control of false positives. We confirmed that linkage disequilibrium (LD) and imperfect expression imputation and pheno-type measurement can lead to false positive epistasis [23,24]. However, across extensive simu-lations, we were only able to inflate the type I error rate in the presence of LD; therefore, we apply a relatively simple yet robust approach to remove findings likely enriched for false posi-tive interaction associations by excluding from analyses pairs of nearby genes and those with correlated imputed expression.

Despite these challenges, we identify genome-wide significant gene-gene interaction associ-ations with problematic alcohol use and generalized anxiety disorder. This is proof-of-princi-ple that the approach will identify novel interactions that can extend our biological understanding of complex traits, and as larger datasets and consortia become available, we anticipate additional epistatic associations will emerge.

Furthermore, when adopting a self-contained gene set-level approach [69], we identified several significantly associated gene sets (Figs 4–5 and S7–S10 Tables). We note that as a self-contained gene set analysis, this is testing a null hypothesis of no pairwise interaction associa-tion of genes within the gene set, rather than an enrichment of association signal relative to the background level of interaction associations (competitive gene set analysis [69]); computa-tional constraints currently limit widespread E-TWIS competitive set analyses, but our follow-up resampling procedure performs such a competitive test, and we found qualitatively similar results, providing a way to verify enrichment of any sets of interest. Identified gene sets of interest include inflammatory and immune system pathways as relating to smoking, alcohol use, GAD and neuroticism; deubiquitination related to alcohol use suggesting the importance of epistasis for posttranslational modification; and multiple, notably glutamatergic, cell signal-ing pathways. Of particular interest, specific relevant cell populations can be identified using E-TWIS, and these include individual neuronal cells as well as glia.

## Limitations

Our exhaustive TWIS study has several notable limitations. First, we applied a linear regres-sion-based statistical definition of epistasis, based on additive SNP effects on expression. This is an additive-by-additive (AxA) definition of epistasis. While computationally efficient, other models of epistasis can affect complex traits [25,26], such as non-linear interactions among gene expression, dominance (D) effects (DxD, AxD), or higher order interactions [21], which are not tested in our framework.

Second, LD leads to correlated tests and correlated predictors, which leads to complications in error control in interaction studies, increasing type I error and false associations of epistasis [23,24]. While standards for type I error correction have been generally accepted in single-SNP GWAS, there is no previous analogous standard for application to interactions. We have addressed this via extensive analyses of power and bias and have taken a conservative approach, removing any nearby pairs of loci and those with correlated imputed gene expres-sion ($|r|>0.05$). This has likely removed true epistatic interactions, in which nearby, linked

genes or intragenic loci interact [10,27,39]. While this prevents identification of physically proximate interactions, it removes a major source of LD-driven false positives [23,24] which we view as necessary.

Third, expression can be influenced by environments and traits themselves. The use of genetically predicted expression reduces the possibility of this kind of confounding [5], but our framework is fundamentally distinct from a traditional SNP-SNP interaction test. TWIS is based on the TWAS framework, and therefore, all limitations of TWAS [41] also apply. For example, related to the second point above, gene pair expression correlation can result from LD between functional variants of each gene, as well as shared functional variants affecting both genes, possibly leading to spurious (non-causal) associations between genes and traits. A second issue in TWAS is heterogeneity among expression reference panels, for instance due to cell type heterogeneity [41]. This is typically assessed using an omnibus test to account for among reference panel heterogeneity [3]. We have limited our analyses to using a single reference panel due to the number of traits and tissues and the number of pairwise tests involved, but incorporating the heterogeneity of reference panels would be a useful avenue of future research.

Fourth, the replication rate for epistasis tests is expected to be substantially lower than for additive tests, due to ascertainment of markers in LD with the causal variants and their chance resampling in independent datasets [10]. Nonetheless, we have applied rigorous replication thresholds, which we acknowledge likely result in higher rates of false negative replication. Combined with the stringent thresholds to remove LD-driven false positives, we are likely underestimating the extent of epistasis throughout the genome in complex traits; larger sample sizes will improve epistasis discovery.

Furthermore, scaling phenotypes in different ways (e.g., logarithmic) will impact the interaction estimates [9,76]. We residualized phenotypes and imputed expression, but the statistical epistasis identified here may be scale-dependent, and further mechanistic studies are required to determine the biological interactions at individual loci. Our analysis represents a computationally demanding, yet initial assessment of interactions throughout the genome.

Finally, assortative mating is expected to lead to correlation (i.e., LD) at functional loci even if they are physically separated [21,77]. We removed correlated loci, those in which assortative mating would be expected to lead to false positives. In this way, we expect assortative mating to not be a large driver of results here, but it is an area of future work worth exploring.

## Conclusions

Epistasis is likely widespread, but the computational challenges of so many pairwise tests have prevented its extensive examination. Here, we present a way forward using predicted gene expression, finding several significant interaction associations and multiple cell types and functional annotations enriched in epistasis affecting complex traits. We anticipate more to be identified as GWAS and expression reference panels continue to grow.

## Methods

### Description of TWIS Approach

We tested all pairs of gene-gene interactions using imputed gene expression after residualizing both the phenotype and expression on multiple covariates. This approach improved computation time while leading to unbiased estimates of the interaction effect. Details of each step are described below.

Scripts to perform TWIS and E-TWIS using publicly available data are available at https://github.com/evanslm/TWIS.

## Gene expression imputation in the prefrontal cortex (PFC) and three orthogonal cross-tissue expression measures

We imputed expression of genes in the PFC using the weights generated by PsychENCODE [32] (14,729 genes) as well as three cross-tissue measures of expression [33] (13,242; 12,521; and 12,032 genes for the three measures). We included the cross-tissue measures of expression (sparse canonical correlation analysis axes [sCCA] 1–3), as integration of data across multiple tissues increases reference sample sizes and improves power [33].

TWAS weights were downloaded from the FUSION website for the PFC and cross-tissue expression measures (http://gusevlab.org/projects/fusion/). For each gene in each tissue, we first created score files of the best performing model weights using the make_score.R script (as outlined and available at the FUSION github site: https://github.com/gusevlab/fusion_ twas). Following standard genotype QC (described below), we next extracted all SNPs in each cohort with non-zero expression weights using plink2 [78], followed by creating the individual-level expression prediction (plink2--score command) for each gene's expression.

## Residualization of imputed expression and phenotypes

In interaction studies, proper control of covariates requires inclusion of all covariate-by-main effect terms [36]. This is critical when possible confounding variables exist. Therefore, we first examined a model for phenotype $y$ in which, for imputed gene expression of two genes, $T_1$ & $T_2$, all main gene expression, expression interaction and covariate-by-gene expression terms were included:

$$y = \mu + \beta_1 T_1 + \beta_2 T_2 + \beta_{int} T_1 T_2 + \sum_{k=1}^{m} \alpha_k cov_k + \sum_{k=1}^{m} \alpha_{k1} cov_k T_1 + \sum_{k=1}^{m} \alpha_{k2} cov_k T_2 + \varepsilon \quad (2)$$

where $\mu$ is the intercept, $\beta_1$ is the effect of expression of gene 1 ($T_1$), $\beta_2$ is the effect of expression of gene 2 ($T_2$), $\beta_{int}$ is their interaction effect, $\alpha_k$ is the effect of the $k^{th}$ covariate ($cov_k$), $\alpha_{k1}$ and $\alpha_{k2}$ are the interaction effects of the $k^{th}$ covariate with $T_1$ and $T_2$, respectively, and $\varepsilon$ is the error term.

Covariates include, depending on availability within each cohort (see Methods), age, sex, genotyping batch, assessment center, socioeconomic variables such as income or education, and the first 10 genome-wide principal component axes. When many covariates are included, such as the large numbers of genotyping batches (106) and assessment centers (22) in the UK Biobank, all $m$ covariates and their interactions with the main gene expression terms rapidly increases to hundreds of additional terms to estimate in the model for each pair of genes. This drastically increased computation time across many pairwise tests, particularly in samples of hundreds of thousands (e.g., the UK Biobank). Even with the reduced number of predictors (at the gene expression level) used here compared to all individual SNPs, all pairwise comparisons reach tens of millions of tests, e.g., ~14,000 genes imputed using the PsychENCODE cortex expression weights [32] results in ~108M pairwise comparisons.

To improve speed, we therefore first residualized both the phenotype and genetically predicted gene expression on all covariates. This approach allowed us to remove covariate effects first, rather than repeatedly estimating them and their interactions for each pairwise test. Residualizing both predictor and response variables leads to unbiased estimates of the gene-gene interaction effect. We extracted the residuals from the following model:

$$x = \mu + \sum_{k=1}^{m} \alpha_k cov_k + \varepsilon \quad (3)$$

where $\mu$, $\alpha_k$, $cov_k$, and $\varepsilon$ are as above, and $x$ is either the phenotype (e.g., height) or the imputed gene expression (e.g., predicted $T_1$). We used *fastLm* in the *RcppArmadillo* [79] R package to

fit the model efficiently for each imputed gene's expression and continuously distributed phenotype, and the *glm* function to fit logistic regressions for each dichotomous phenotype. Residualized imputed expression and phenotype data were then merged into a single data frame.

## Exhaustive, all pairs gene-gene interaction TWIS

Within each cohort, we then performed an exhaustive (all pairs) TWIS within each tissue for each trait using the following model:

$$y_{resid} = \mu + \beta_1 T_{1resid} + \beta_2 T_{2resid} + \beta_{int} T_{1resid} * T_{2resid} + \varepsilon \tag{4}$$

where $y_{resid}$ indicates the residuals of phenotype $y$ and $T_{1resid}$ and $T_{2resid}$ are the residuals of predicted gene expression of $T_1$ and $T_2$, respectively, from eq 3. We estimated $\mu$, $\beta_1$, $\beta_2$, and $\beta_{int}$ using fastLm in R. For each tissue and trait within each cohort, this amounted to $gp$ = 108,464,356; 87,668,661; 78,381,460; and 72,379,496 pairwise tests in the PFC and three cross-tissue expression measures, respectively, or 346,892,973 total pairwise tests for each trait in each cohort.

To expedite this step, we parallelized these tests across multiple compute nodes using the RMACC Summit Supercomputer at CU Boulder. For each combination of tissue, trait, and cohort, we split the total tests into 1000 chunks, each of which was distributed to independent compute nodes. Each chunk therefore performed $gp/1000$ pairwise tests, which were indexed as the tests between pair ($a[k]$, $a[i+k+d*(d+1)/2-m]$), where $n$ = number of total genes, $m = n*(n-1)/2$, $y = m-i$, $i$ is the $i^{th}$ chunk out of 1000, $d$ = 1+floor((($8*y+1$)^0.5−1) / 2), and $k = n-d$. This uniquely tested each pair only once, while distributing the computation to as many compute nodes as available on the supercomputer. Within each chunk, we further parallelized eq 4 to multiple available CPUs using the *foreach* R library [80].

## Discovery, replication, and meta-analysis

We treated the UK Biobank as the discovery sample, and meta-analyzed results from the remaining cohorts for each phenotype as an independent replication sample. For meta-analysis, we applied the sample-size weighted approach of METAL [42]. We applied this rather than a traditional inverse-variance weighted meta-analysis because in several cases, the phenotypes in each cohort were approximate comparisons (e.g., "psychiatric disorder" based on ICD-9 & -10 codes (GERA, UK Biobank) vs. self-reported and DSM-V diagnoses of multiple disorders (ARIC, NESARC-III) and because the predictors and phenotypes were residualized on covariates prior to our TWIS, making SE-based meta-analysis inappropriate.

A full description of power and type-I error rates is in *Determining Alpha* and *Tests of Power and Biases* below. Based on those findings, we applied a significance threshold of $\alpha$ = 5.86e-10. When pairs of genes are unlinked, this is the approximate 5[th] percentile of minimum $p$-values from exhaustive genome-wide gene-gene tests under the null (see below). This is also very close to the Bonferroni correction threshold for all pairs of genes across the genome (i.e., ~0.05/choose(20000,2)). Based on those findings and tests of biases described below, we restrict all subsequent analyses to pairs of genes whose physical position midpoints are greater than 1Mb apart and whose imputed expression is uncorrelated ($|r| < 0.05$), because linked and correlated pairs of genes lead to high rates of false positives. In our independent replication dataset, at pairs passing discovery significance, we applied a nominal $p<0.05$ as evidence of replication. Finally, we meta-analyzed all cohorts together (UKB+ replication cohorts). The complete meta-analysis results were utilized in subsequent gene set enrichment tests.

## Sample QC, stratification, PCA and relatedness

All cohorts (S1 Note) included SNP array and/or imputed genome-wide SNP data. Genotype quality control (QC) of the array data included genotype missingness, Hardy-Weinberg Equilibrium tests, and minor allele frequency (MAF) using plink2 (command:—geno 0.05—hwe 0.00000001—maf 0.01). For cohorts without imputed data, we utilized the Michigan Imputation Server to impute array data to the Haplotype Reference Panel [81,82] after QC. Following imputation, then applied additional QC imputation metrics using plink2 (command:--extract-if-info R2 '> = ' 0.9--maf 0.0001--hwe 0.00000001--geno 0.01--mind 0.01).

Within each cohort, we identified a set of unrelated and relatively unstratified individuals matching (in terms of principal components analysis [PCA] axes) the expression reference panels, which are primarily European ancestry individuals. To reduce stratification effects and because expression imputation relies on sufficient matching of LD patterns between the target and reference panels [83], we restricted our analyses to individuals of European ancestry, as that was both the largest relatively genetically homogeneous sample available across all cohorts and because the expression reference data were primarily derived from European ancestry individuals. We first used HapMap3 positions in the 1000 Genomes (1KGv3) [84] reference panel to generate PCA loadings of the first 10 axes using flashpca [85]. We then extracted these same HapMap3 positions from each study cohort and projected them onto the 1KGv3 PC axes using flashpca. We then identified all individuals within +/-5 standard deviations of the 1KGv3 EUR population mean on each of the first four PCs, matching the approach applied by GSCAN [62] across many cohorts, thus identifying a relatively unstratified set of individuals with LD patterns roughly matching those of the expression reference panels available.

We retained unrelated individuals using GCTA [86] within each cohort after applying a pairwise relatedness cutoff of 0.05 using MAF- and LD-pruned SNPs (plink2--maf 0.01--indep-pairwise 50 5 0.2). See S1 Table for final sample sizes for each cohort and each phenotype.

## Tests of power and biases

We performed a series of simulations to estimate power to detect interactions in the context of imperfect expression imputation across a range of epistasis effect sizes, define the appropriate alpha for genome-wide multiple test correction in the context of many millions of individual tests, and assess the role of LD in influencing interaction tests.

## Assessment of power in the context of expression prediction error

Genetically based expression prediction is imperfect (i.e., prediction $r^2 <$ expression $h^2_{SNP} < 1$ S1 Fig). This is a function both of the heritability of the trait [87] as well as sampling variance from finite (often small) expression reference panels [3–5]. To assess how such imperfect expression prediction impacted the power to detect gene-gene (GxG) expression interactions, we performed a set of Monte Carlo simulations (each 5,000 replicates) while varying the sample size (N = 5000, 10000, 15000, 25000, 40000, 50000, 75000, 100000, 150000, 200000, 250000, 500000), the proportion of the phenotypic variance truly explained (PVE) by the interaction (PVE = 0, 0.0001, 0.00025, 0.0005, 0.001, 0.005), and incorporating prediction error of the gene expression values by drawing randomly from the observed distribution of imputation accuracy (S1 Fig). We simulated gene expression values (the predictors in our model) from standard normal distributions, then generated phenotypes as a function of main and interaction expression effects and random noise, based on the set PVE. We then added error to the predictor expression values by drawing random noise from a $\sim N(0, \sigma^2_{resid})$, where $\sigma^2_{resid}$ was equal to one minus the observed prediction accuracy of a value randomly drawn from the

distribution in S1 Fig. We performed these simulations with and without the added prediction error to assess its influence on bias and power.

As expected, decreased PVE and added expression error both decreased the power to detect significant interactions (S2 Fig). Note that when PVE = 0, roughly 5% of tests were significant when using alpha = 0.05 (and 0% with more stringent thresholds), indicating a well-calibrated interaction test statistic under these simulated conditions.

## Assessment of power using actual predicted gene expression

We next used UK Biobank data, with genome-wide predicted gene expression, to incorporate real predicted expression data into our simulations. We used predicted sCCA1 expression data, and excluded individuals with relatedness > 0.05 (e.g., a sample similar to that used when testing epistasis effects on height). We randomly selected 5,000 pairs of genes from throughout the genome, and from the imputed expression data, simulated phenotypes as described above. We then added random noise to the imputed expression predictors, based on the estimated prediction accuracy (S1 Fig) for each gene in each pair. Again, power declined when error (due to imperfect expression prediction models) was added to the expression values used in the regressions (S3 Fig). Power was also decreased relative to the simulations described above (S2 Fig). Note that when PVE = 0, roughly 5% of tests were significant when using alpha = 0.05 (and 0% with other thresholds), indicating a well-calibrated interaction test statistic when incorporating data derived from real imputed expression data within a large biobank sample.

## Assessment of power using pairs of physically proximate genes when local SNPs are in LD

In the presence of imputation error, LD leads to an inflated false positive rate. We confirmed that, similar to recent reports [23,24], this is due to binomially distributed predictors (i.e., true expression abundance when genetically based) with normally distributed error added (either from imperfect expression imputation or from random error) through a series of simulations varying LD, physical proximity and the distribution of the predictors (binomially distributed or normally distributed gene expression levels). We found evidence for this inflation only in the presence of LD. We next describe the two analyses we performed to conclude this.

Variation within nearby genes is expected to be correlated due to LD of SNPs, and we expected that this could inflate test statistics, leading to false positives when comparing physically proximate genes based on other studies [23,24]. To understand how LD impacts the test statistics, we therefore performed tests identical to those described above, but randomly chose only pairs of genes that were physically, immediately next to one another, thereby building into the simulations the desired physical proximity and underlying LD among causal variants. In these simulations (S4 Fig), power to detect true effects was slightly reduced relative to when pairs were randomly selected throughout the genome, but when prediction error was added to the expression values, we observed inflation of the Type I Error rate. When PVE = 0, at the largest samples simulated, ~40%, 7.5%, and 5.5% of tests were significant at alpha = 0.05, 5e-8, and 2.5e-10.

To confirm LD as the cause of this, we simulated pairs of gene expression data from either a standard normal distribution ($\sim N(0,1)$) or from a simple PGS (the sum of the minor alleles) of varying polygenicity (2, 10, 20, 50 or 100 SNPs per gene) derived from binomially distributed genotype data. We then generated phenotypes from the main effects of the simulated gene expression but without a true interaction. For each simulation, we tested the regression model interaction term, either using the simulated PGS (representing the simulated expression of each gene) or simulating imperfect expression prediction by adding normally distributed noise to the PGS (S5 Fig). When using the true PGS as the predictor (no predictor error), the

interaction tests are well calibrated (Type I error rate ~0.05 when applying alpha = 0.05) regardless of trait architecture or LD. When SNPs affecting gene expression are not in linkage disequilibrium, the interaction tests are also well calibrated. However, if the SNPs affecting expression are in LD (such as would occur for perfectly correlated PGSs of nearby genes), type I error rates can become strongly inflated in the presence of imperfect expression. When using expression with added error (to mimic imperfectly predicted expression data), the false positive rate becomes much greater if the expression is predicted from a PGS generated from simulated, binomially distributed SNPs. The effect is greatest for a PGS derived from a few SNPs with poor prediction accuracy (high error variance added to the predictor), and declines as the expression polygenicity increases or the prediction accuracy improves. When estimated expression was derived from a standard normal distribution, the type I error rates were never inflated. This appears to be due to the combination of a binomially distributed predictor with added error variance, a situation that has been observed previously [23,24].

These results suggest that tests of nearby genes (those with SNPs as predictors in LD) have inflated type I error rate and should be treated with caution. Gene pairs physically or with unlinked SNPs affecting them are unaffected, and the type I error rate is well calibrated.

## Assessment of expression-based interaction tests when causal variant effects do not operate via expression

We assessed the impact of true genetic interactions that are not mediated via expression effects on the phenotypes. Predicted cross-tissue or tissue-specific expression data are essentially local PGSs, built from SNPs within localized physical windows. If there are true causal variants (CVs) that impact the phenotype directly (*not* through effects on gene expression) and are linked to SNPs that predict gene expression, it is possible that one could identify significant GxG expression PGS-based associations due to LD, when in fact no *expression-based* interactions truly influence the phenotype.

We tested this by simulating SNP-by-SNP interaction effects on phenotypes, then testing models of either SNP-SNP interactions or expression PGS gene-gene interactions. In these simulations, there is a true genetic interaction effect via SNPs, but the phenotype is unaffected by genetic-based expression. We included two different scenarios to confirm that LD between the truly functional SNPs and the rest of the SNPs that contribute to the genetically predicted expression is what drives the TWIS associations, using either the SNPs with the locally maximal LD score or the SNPs with the locally minimal LD score as the truly interacting SNPs.

Consistent with expectations, we found this results in false positive associations of gene expression epistasis, which reflects the expression PGS tagging of true causal interactions (S6 Fig). Furthermore, the larger the LD scores of the interacting SNPs, the higher the false positive rate of TWIS associations. We note that this is a false positive in the sense that there are no *expression-mediated* interactions, but there is a true genetic interaction in these scenarios, so such false positives may still be of interest.

## Determining alpha

The study-wide alpha based on a Bonferroni correction is approximately $0.05/C_2^{20,000} \cong 2.5e\text{-}10$ for a single trait and tissue expression combination assuming 20,000 genes in the genome, but these tests are not independent due to LD and their pairwise nature. Furthermore, the influence of LD, described above, clearly leads to inflated false positive rates. We estimated an appropriate genome-wide multiple test correction threshold by applying a similar simulation-based approach as has been used for univariate GWAS [88]. We simulated 100 independent genome-wide TWIS studies, each with 13,224 genes and 87,430,476 pairwise tests of epistasis

(8,743,047,600 total tests) using the imputed sCCA1 expression in unrelated individuals from the UK Biobank, matching the sample size with height data (N = 328,745). In each of the 100 datasets, we simulated phenotypes for each pair of gene-gene interaction tests, in which the genes had true main effects but no interaction effects, then estimated the interaction effect $p$-value using the approach described above. We identified in each of the 100 simulated TWIS studies the minimum $p$-value, then identified the 5[th] percentile of these 100 minimum $p$-values as the appropriate genome-wide alpha. However, because LD varies throughout the genome and is expected to inflate false positive rates, we split this analysis into tests in which both genes are found on the same vs. different chromosomes, as a proxy for pairs in which SNPs are possibly in LD vs. those not in LD. For 60 of these simulated TWIS studies, we further assessed whether the distribution of interaction $p$-values are drawn from an approximate cumulative $t$ distribution across a range of pairwise expression correlations using Kolmogorov-Smirnov (K-S) tests, implemented in R. We found that the 5[th] percentile of minimum $p$-values across the 100 simulated TWIS datasets from gene pairs on the same chromosome is 1.22e-20, reflecting the test statistic inflation due to LD between SNPs nearby the genes noted above, while the 5[th] percentile of minimum p-values from gene pairs on different chromosome is 5.86e-10, very similar to the alpha when using a Bonferroni correction (S7 Fig). While predicted expression of all pairs of genes on different chromosomes were generally uncorrelated (most $|r|$<0.1) and the $p$-value distribution was not different from the expected cumulative $t$ distribution, pairs on the same chromosomes had a range of pairwise expression correlation, and the distribution of $p$-values was increasingly dissimilar from expected at stronger pairwise expression correlations (S8 Fig). Notably, across the 60 simulated datasets, the K-S test was not significant (almost all $p$>0.05) when, on the same chromosome, pairwise gene expression $|r|$<0.1, giving a threshold of pairwise expression correlation due to local LD, above which false positives are likely, but below which test statistics are reasonably well-calibrated. We therefore use a genome-wide, exhaustive TWIS corrected significance threshold of $p$<5.86e-10, while conservatively also excluding any pairs of genes whose imputed expression $|r|$>0.05 in the discovery sample (UK Biobank sample) and those pairs within 1MB.

## Enrichment-TWIS (E-TWIS)

We estimated enrichment of interaction associations within gene sets, rather than individual pairs of genes. We assessed the strength of interaction associations among genes within gene sets using a network analysis approach to determine the connectedness of all pairs of genes within *a priori* defined gene sets of interest, including multiple functional pathways and networks. Similar to network connectivity [68], our measure summed the squared, meta-analyzed, pairwise interaction association $Z$-scores of all $m$ pairs of $n$ genes within each pathway or gene set, which was $\chi^2_{df = m}$-distributed. To confirm that this approach produces appropriate $p$-values of gene set TWIS association enrichment, we performed simulations to estimate the distribution of gene set $\chi^2$ statistics under the null of no interaction association for several gene sets of varying size. These confirmed that our test statistic was roughly $\chi^2_{df = m}$-distributed for small ($n$<150) gene sets but was anti-conservative for very large gene sets (S38 Fig). In these cases, we employed a secondary strategy, in which we randomly resampled $n$ genes 1000 times, approximating the length and number of variants per gene in the target dataset, and averaged their $m$ pairwise, squared TWIS $Z$-scores to estimate an empirical enrichment $p$-value. We confirmed similar findings to the $\chi^2_m$ test (S39–S40 Figs), noting that resampling represents a competitive test (*sensu* [69]) accounting for background heritability throughout the genome via resampling random genes; therefore, annotated gene sets identified via random resampling are concluded to be enriched relative to background epistatic interactions.

We advocate an approach of efficiently testing many gene sets via $\chi^2$ tests and resampling to confirm large gene set enrichment or to test sets of particular interest.

We tested a broad range of gene sets, including the weighted gene coexpression network analysis (WGCNA) modules in the PFC identified by Gandal et al.[32,70] and multiple sets from the Molecular Signatures Database (MsigDB) [71]. The latter included hallmark gene sets; c2 canonical curated genesets from Biocarta, KEGG, and Reactome pathways; c3 regulatory target gene sets; c7 ImmuneSigDB gene sets; and c8 cell type signature gene sets. After excluding sets with fewer than 10 genes, we tested a total of 7,911–8,012 sets per trait and expression tissue and applied FDR≤0.05 multiple test correction. We then used the same approach to assess interaction association enrichment in genes specifically expressed within individual cell types within multiple brain regions and subsets of those genes that are intolerant to protein-truncating mutations (defined in [72]).

## Number of interactions per gene

To examine the distribution of interaction frequency per gene, we applied a nominal significance threshold of interaction $p \leq$ 1e-5. We then evaluated the number of interactions each gene was involved in by plotting the distribution. As demonstrated by our power simulations, we are underpowered to detect strict Bonferroni-significant interactions, but as demonstrated by our gene set enrichment analyses, there is a signal of interaction associations within tests that do not reach strict significance, which is why we used a nominal $p \leq$ 1e-5 threshold.

## TWIS vs. TWAS comparison using UK Biobank data

We assessed whether genes identified in TWIS would have been identified using single gene TWAS [3], as it has been hypothesized the effect sizes of a locus could be muted when analyzed individually if the gene's effect depends on an interaction with another gene [7]. We restricted our analysis to genes within pairs of suggestive ($p \leq$ 1e-5) interaction associations in the combined meta-analysis, across any phenotype and trait combination. We used the residualized UK Biobank data and applied a $p \leq$ 1e-5 suggestive significance criteria. Using these results, we compared the interaction effect sizes from TWIS for each gene with its TWAS-estimated effect size to test whether genes with larger interactions have smaller effect sizes estimated in a single-locus model.

## SNPxSNP Interactions Follow-up analyses

Five pairs of gene interactions were significantly associated in the final meta-analysis (Table 1). We therefore followed up these TWIS associations with all pairwise SNPxSNP interaction associations with the same set of traits. We extracted all SNPs +/- 500KB of the gene transcription start site (matching the FUSION TWAS weight calculation windows [4]), residualized the phenotype and SNP genotypes on the same covariates as in TWIS and performed all SNPxSNP interaction tests for each pair of genes found in Table 1. We performed these analyses in all cohorts with trait data, then meta-analyzed the interaction associations as described above. Full results of all SNPxSNP interaction tests as well as the full meta-analyzed TWIS results are available on Dryad [89].

**Dryad DOI.** https://doi.org/10.5061/dryad.866t1g1tw

## Supporting information

**S1 Note. Cohort and phenotype descriptions.**
(DOCX)

**S1 Fig. Histogram of the best model (i.e., lowest p-value) from FUSION output of the cross-tissue expression prediction models (first sCCA axis).**
(TIFF)

**S2 Fig. Power to detect significant interactions at two significance thresholds across varying sample sizes and true proportions of phenotypic variance explained (PVE) by the interaction.** Left, without incorporating expression prediction error into the simulation. Right, incorporating random error for each predicted gene expression based on the distribution of observed prediction accuracies of the best model in S1 Fig. Main effect sizes for the two expression predictors (T1 & T2) are shown above each plot; varying these had minimal effect on the interaction test power.
(TIFF)

**S3 Fig. Power to detect significant interactions at two significance thresholds across varying sample sizes and true proportions of phenotypic variance explained (PVE) by the interaction when using predicted sCCA1 expression within the UK Biobank in unrelated individuals using pairs of genes randomly selected throughout the genome.** Left, without incorporating expression prediction error into the simulation. Right, incorporating random error for each predicted gene expression based on the observed prediction accuracies of the best model in S1 Fig. Main effect sizes for the two expression predictors (T1 & T2) are shown above each plot; varying these had minimal effect on the interaction test power.
(TIFF)

**S4 Fig. Power to detect significant interactions at two significance thresholds across varying sample sizes and true proportions of phenotypic variance explained (PVE) by the interaction when using predicted sCCA1 expression within the UK Biobank in unrelated individuals using pairs of genes that were immediately next to one another in the genome.** Left, without incorporating expression prediction error into the simulation. Right, incorporating random error for each predicted gene expression based on the observed prediction accuracies of the best model in S1 Fig.
(TIFF)

**S5 Fig. The proportion of significant tests (alpha = 0.05) when simulating gene expression from linked (correlation of expression or gene LD = 1) or unlinked (= 0) data, either from a standard normal distribution ($\sim N(0,1)$) or from a simple PRS of varying polygenicity.** Simulated phenotypes included main effects of gene expression (based on varying polygenicity), but did not include gene expression interaction effects. When using truly normally distributed gene expression values in the regression (top), the test statistic is well calibrated (i.e., Type I error rate $\sim$alpha), regardless of whether additional variance is added and whether estimated (i.e., imperfectly predicted) expression data are used. However, when the true expression data is generated from binomially distributed SNPs, using an imperfectly predicted PGS results in inflation of the Type I error rate, proportional to how poorly the PGS predicts expression, i.e., with increasing error variance added to the predictor. Note that this does not occur when the true observed expression is used, even if binomially distributed. The effect is greatest for a PGS using a single SNP, and weakens as the expression becomes more polygenic.
(TIFF)

**S6 Fig. Simulations of causal SNPxSNP effects on the phenotype, tested either using SNPxSNP interactions (top) or imputed expression gene-gene interactions (bottom), when varying the LD (based on the LD score) of the causal SNPs.** False positives increase when the

SNPxSNP CVs have high LD scores than low LD scores, to the extent that the true effect is driven by SNP-SNP interactions, not expression-expression interactions. This results from LD between the causal SNPs and those used in the expression imputation.
(TIFF)

**S7 Fig. Distribution of the minimum interaction p-value for each of 100 simulated TWIS studies (each observation represents the minimum p-value of ~87M pairwise interaction tests across the genome), separated by whether the pair of genes is on the same (top) or different chromosomes (bottom).** Red dashed line represents the 5[th] percentile of the minimum p-values. Note the x-axis scale differs between the two panels.
(TIFF)

**S8 Fig. Violin plots of K-S test p-value testing whether the distribution of interaction test statistics is t-distributed across 40 whole genome TWIS study replications, depending on the pairwise imputed expression correlation between the gene pairs, and separated by whether the pair of genes is on the same (top) or different chromosomes (bottom).** NA indicates no pairs of genes were found within that bin of pairwise imputed expression correlation. Blue dots are the (jittered) individual K-S test p-values for an entire simulated TWIS study.
(TIFF)

**S9 Fig. Boulder plot of BMI interaction association p-values using imputed transcription.** Shown are the results from the final meta-analysis of all data. Black lines connect pairs that surpassed p<2.5e-10 in the discovery cohort (UKB), blue lines connect pairs of loci with nominally significant interaction (p<0.05) in the replication cohort, and gray lines connect pairs of genes with p<2.5e-10 in the final meta-analysis.
(TIFF)

**S10 Fig. Boulder plot of cAUDIT interaction association p-values using imputed transcription.** Shown are the results from the final meta-analysis of all data. Black lines connect pairs that surpassed p<2.5e-10 in the discovery cohort (UKB), blue lines connect pairs of loci with nominally significant interaction (p<0.05) in the replication cohort, and gray lines connect pairs of genes with p<2.5e-10 in the final meta-analysis.
(TIFF)

**S11 Fig. Boulder plot of CPD (heavy CPD> = 20 vs light CPD< = 10) interaction association p-values using imputed transcription.** Shown are the results from the final meta-analysis of all data. Black lines connect pairs that surpassed p<2.5e-10 in the discovery cohort (UKB), blue lines connect pairs of loci with nominally significant interaction (p<0.05) in the replication cohort, and gray lines connect pairs of genes with p<2.5e-10 in the final meta-analysis.
(TIFF)

**S12 Fig. Boulder plot of DPW interaction association p-values using imputed transcription.** Shown are the results from the final meta-analysis of all data. Black lines connect pairs that surpassed p<2.5e-10 in the discovery cohort (UKB), blue lines connect pairs of loci with nominally significant interaction (p<0.05) in the replication cohort, and gray lines connect pairs of genes with p<2.5e-10 in the final meta-analysis.
(TIFF)

**S13 Fig. Boulder plot of GAD interaction association p-values using imputed transcription.** Shown are the results from the final meta-analysis of all data. Black lines connect pairs that surpassed p<2.5e-10 in the discovery cohort (UKB), green lines connect pairs of loci with

significant (q<0.05) in the replication cohort, and gray lines connect pairs of genes with p<2.5e-10 in the final meta-analysis.
(TIFF)

**S14 Fig. Boulder plot of height interaction association p-values using imputed transcription.** Shown are the results from the final meta-analysis of all data. Black lines connect pairs that surpassed p<2.5e-10 in the discovery cohort (UKB), blue lines connect pairs of loci with nominally significant interaction (p<0.05) in the replication cohort, and gray lines connect pairs of genes with p<2.5e-10 in the final meta-analysis.
(TIFF)

**S15 Fig. Boulder plot of MDD interaction association p-values using imputed transcription.** Shown are the results from the final meta-analysis of all data. Black lines connect pairs that surpassed p<2.5e-10 in the discovery cohort (UKB), blue lines connect pairs of loci with nominally significant interaction (p<0.05) in the replication cohort, and gray lines connect pairs of genes with p<2.5e-10 in the final meta-analysis.
(TIFF)

**S16 Fig. Boulder plot of neuroticism interaction association p-values using imputed transcription.** Shown are the results from the final meta-analysis of all data. Black lines connect pairs that surpassed p<2.5e-10 in the discovery cohort (UKB), blue lines connect pairs of loci with nominally significant interaction (p<0.05) in the replication cohort, and gray lines connect pairs of genes with p<2.5e-10 in the final meta-analysis.
(TIFF)

**S17 Fig. Boulder plot of pAUDIT interaction association p-values using imputed transcription.** Shown are the results from the final meta-analysis of all data. Black lines connect pairs that surpassed p<2.5e-10 in the discovery cohort (UKB), blue lines connect pairs of loci with nominally significant interaction (p<0.05) in the replication cohort, and gray lines connect pairs of genes with p<2.5e-10 in the final meta-analysis.
(TIFF)

**S18 Fig. Boulder plot of psychiatric interaction association p-values using imputed transcription.** Shown are the results from the final meta-analysis of all data. Black lines connect pairs that surpassed p<2.5e-10 in the discovery cohort (UKB), blue lines connect pairs of loci with nominally significant interaction (p<0.05) in the replication cohort, and gray lines connect pairs of genes with p<2.5e-10 in the final meta-analysis.
(TIFF)

**S19 Fig. Boulder plot of smoking cessation (SC) interaction association p-values using imputed transcription.** Shown are the results from the final meta-analysis of all data. Black lines connect pairs that surpassed p<2.5e-10 in the discovery cohort (UKB), blue lines connect pairs of loci with nominally significant interaction (p<0.05) in the replication cohort, and gray lines connect pairs of genes with p<2.5e-10 in the final meta-analysis.
(TIFF)

**S20 Fig. Boulder plot of smoking interaction (SI) interaction association p-values using imputed transcription.** Shown are the results from the final meta-analysis of all data. Black lines connect pairs that surpassed p<2.5e-10 in the discovery cohort (UKB), blue lines connect pairs of loci with nominally significant interaction (p<0.05) in the replication cohort, and gray lines connect pairs of genes with p<2.5e-10 in the final meta-analysis.
(TIFF)

**S21 Fig. GAD (case = 1, control = 0, jittered for visualization) plotted against imputed expression of x ENSG00000086475.14 for values of ENSG00000166912.16 either above or below the median (high or low, respectively), imputed using sCCA3 cross tissue weights.** Studies are indicated in title of each panel. Fitted logistic regressions are shown by dashed line.
(TIFF)

**S22 Fig. pAUDIT plotted (1 or 0, jittered) against imputed expression of ENSG00000115596 for values of ENSG00000126583 either above or below the median (high or low, respectively), imputed using PFC expression weights.** Studies are indicated in title of each panel. Fitted logistic regressions are shown by dashed line.
(TIFF)

**S23 Fig. pAUDIT plotted (1 or 0, jittered) against imputed expression of ENSG00000135525 for values of ENSG00000126583 either above or below the median (high or low, respectively), imputed using PFC expression weights.** Studies are indicated in title of each panel. Fitted logistic regressions are shown by dashed line.
(TIFF)

**S24 Fig. pAUDIT plotted (1 or 0, jittered) against imputed expression of ENSG00000174938 for values of ENSG00000126583 either above or below the median (high or low, respectively), imputed using PFC expression weights.** Studies are indicated in title of each panel. Fitted logistic regressions are shown by dashed line.
(TIFF)

**S25 Fig. pAUDIT plotted (1 or 0, jittered) against imputed expression of ENSG00000115112 for values of ENSG00000166451 either above or below the median (high or low, respectively), imputed using PFC expression weights.** Studies are indicated in title of each panel. Fitted logistic regressions are shown by dashed line.
(TIFF)

**S26 Fig. Distribution of number of interaction associations for each gene with at least one suggestive (p<1e-5) interaction with BMI.**
(TIFF)

**S27 Fig. Distribution of number of interaction associations for each gene with at least one suggestive (p<1e-5) interaction with cAUDIT.**
(TIFF)

**S28 Fig. Distribution of number of interaction associations for each gene with at least one suggestive (p<1e-5) interaction with CPD (high vs. low use).**
(TIFF)

**S29 Fig. Distribution of number of interaction associations for each gene with at least one suggestive (p<1e-5) interaction with DPW.**
(TIFF)

**S30 Fig. Distribution of number of interaction associations for each gene with at least one suggestive (p<1e-5) interaction with GAD.**
(TIFF)

**S31 Fig. Distribution of number of interaction associations for each gene with at least one suggestive (p<1e-5) interaction with height.**
(TIFF)

**S32 Fig. Distribution of number of interaction associations for each gene with at least one suggestive (p<1e-5) interaction with MDD.**
(TIFF)

**S33 Fig. Distribution of number of interaction associations for each gene with at least one suggestive (p<1e-5) interaction with neuroticism.**
(TIFF)

**S34 Fig. Distribution of number of interaction associations for each gene with at least one suggestive (p<1e-5) interaction with pAUDIT.**
(TIFF)

**S35 Fig. Distribution of number of interaction associations for each gene with at least one suggestive (p<1e-5) interaction with psychiatric.**
(TIFF)

**S36 Fig. Distribution of number of interaction associations for each gene with at least one suggestive (p<1e-5) interaction with smoking cessation (SC).**
(TIFF)

**S37 Fig. Distribution of number of interaction associations for each gene with at least one suggestive (p<1e-5) interaction with smoking initiation (SI).**
(TIFF)

**S38 Fig. Distribution of summed, squared interaction Z-scores for 1000 simulated phenotypes under a null of no epistasis but including main effects for pAUDIT using cortex imputed expression for 6 different gene sets of varying size.** Observed value shown by blue line, while the red line represents the $X^2$ density for the same df. These simulations show that for most gene sets, a standard $X^2$ test is appropriate, but can be anti-conservative for large gene sets, likely when there is a true signal (e.g., gandal_wgcna_CD3 set).
(TIFF)

**S39 Fig. Distribution of mean squared interaction Z-scores for 1000 resampled gene sets for pAUDIT using cortex imputed expression for the same 6 different gene sets of varying size in S32 Fig.** Observed value shown by blue line. These simulations show that for most gene sets, a random resampling approach recapitulates the results of a standard $X^2$ test.
(TIFF)

**S40 Fig. Comparison of p-values from a standard *X2m* test vs.** 1000 randomly resampled gene sets of the same size across a range of observed $X^2_m$ p-values. Note that in the bottom panel, all cases where the resampled *p*-value was <1/1000 (i.e., none of the resampled sets had larger mean $Z^2$ than the observed), -log10(*p*) was set to 4.
(TIFF)

**S1 Table. Sample sizes of unrelated individuals and merging with imputed expression and covariates, representing individuals with complete phenotypic, imputed transcript, and covariate information.** UK Biobank used as discovery dataset. All others meta-analyzed as an independent replication dataset. Finally, all cohorts were meta-analyzed.
(XLSX)

**S2 Table. Simulation results of rates of false positives (a = 0.05) under different trait & predictor residualization approaches, compared to a full model, under a null in which there is no gene-gene interaction effect.** Compared models include the approach used in the main analyses, 'Residualize Expression', in which the imputed expression and trait values were

residualized on all covariates prior to running the test (yresid = T1resid + T2resid + T1resid*-T2resid), and the "Residualize Expression and GxG Term", in which the trait, imputed expression and imputed expression interaction terms were all residualized prior to running the test (yresid = T1resid + T2resid + resid(T1*T2)). "Full Model" includes the observed trait values, the imputed expression and covariate main effects, the T1*T2 and all expression*covariate terms. Simulations used a sample size of 5,000 and 2,000 replicates. Residualization of the trait and the imputed expression never leads to systematically higher rates of false positives than the full model, but residualizing the T1*T2 term separately leads to high rates of false positives when covariates and imputed gene expression values are correlated.
(XLSX)

**S3 Table. Table of significance thresholds, with description of context in which they were or were not applied.**
(XLSX)

**S4 Table. All pairs with significant interactions at either discovery, replication, and/or the final meta-analysis stage.**
(XLSX)

**S5 Table. Counts of number of interaction associations for each gene with at least one suggestive (p<1e-5) interaction with each trait in each tissue.**
(XLSX)

**S6 Table. Comparison of the number of unique genes identified by TWIS and TWAS using the UK Biobank and applying the suggestive significance threshold of p<1e-5.**
(XLSX)

**S7 Table. Gene set association statistics for each trait using expression in each tissue.**
Included are the test statistics before and after removing pairs of genes that are either nearby (<1Mb apart) or have correlated imputed expression (|r|>0.05).
(XLSX)

**S8 Table. Set associations for gene sets with at least one significant (FDR<5%) association after removing pairs of nearby or correlated genes.** Gene set names include the GSEA MsigDB category (e.g., c3.tft.v7.5.1) in addition to the specific gene set.
(XLSX)

**S9 Table. Neuronal cell type gene set association statistics for each trait using expression in each tissue.** Included are the test statistics before and after removing pairs of genes that are either nearby (<1Mb apart) or have correlated imputed expression (|r|>0.05).
(XLSX)

**S10 Table. Set associations for gene sets with at least one significant (FDR<5%) association after removing pairs of nearby or correlated genes.** Gene set names include the GSEA MsigDB category (e.g., c3.tft.v7.5.1) in addition to the specific gene set.
(XLSX)

## Acknowledgments

We thank the participants of the UK Biobank, NESARC-III, Genes for Good, ARIC, and GERA, and we thank the studies and their administrators. This research has been conducted using the UK Biobank Resource (application number 1665).

This work utilized the Summit supercomputer, which is supported by the National Science Foundation (awards ACI-1532235 and ACI-1532236), the University of Colorado Boulder, and Colorado State University. The Summit supercomputer is a joint effort of the University of Colorado Boulder and Colorado State University. This work utilized the Blanca condo computing resource at the University of Colorado Boulder. Blanca is jointly funded by computing users and the University of Colorado Boulder. Data storage supported by the University of Colorado Boulder 'PetaLibrary'. In particular, we thank Andrew Monaghan of CU Research Computing.

GERA Acknowledgement: Data came from a grant, the Resource for Genetic Epidemiology Research in Adult Health and Aging (RC2AG033067; Schaefer and Risch, PIs) awarded to the Kaiser Permanente Research Program on Genes, Environment, and Health (RPGEH) and the UCSF Institute for Human Genetics. The RPGEH was supported by grants from the Robert Wood Johnson Foundation, the Wayne and Gladys Valley Foundation, the Ellison Medical Foundation, Kaiser Permanente Northern California, and the Kaiser Permanente National and Northern California.

Community Benefit Programs. The RPGEH and the Resource for Genetic Epidemiology Research in Adult Health and Aging are described in the following publication, Schaefer C, et al., The Kaiser Permanente Research Program on Genes, Environment and Health: Development of a Research Resource in a Multi-Ethnic Health Plan with Electronic Medical Records, In preparation, 2013 [90].

The origin of the data is described in detail in Hoffmann et al. [91]. Funding support was provided by the National Institutes of Health, National Heart, Lung, and Blood Institute (NHLBI) grant R01 HL128782. We thank our collaborators who created and maintain the datasets used from KAISER and UCSF (phs000788.v1.p2). We are grateful to Kaiser Permanente members, whose participation in the research program makes this genotyping project possible.

## Author Contributions

**Conceptualization:** Luke M. Evans, Jerry A. Stitzel, Marissa A. Ehringer, Charles A. Hoeffer.

**Data curation:** Luke M. Evans, Travis J. Mize, Maizy S. Brasher.

**Formal analysis:** Luke M. Evans, Christopher H. Arehart, Travis J. Mize, Maizy S. Brasher, Marissa A. Ehringer.

**Funding acquisition:** Luke M. Evans, Jerry A. Stitzel, Charles A. Hoeffer.

**Investigation:** Luke M. Evans.

**Methodology:** Luke M. Evans, Christopher H. Arehart, Andrew D. Grotzinger.

**Project administration:** Luke M. Evans.

**Resources:** Luke M. Evans.

**Software:** Luke M. Evans, Christopher H. Arehart.

**Supervision:** Luke M. Evans.

**Visualization:** Luke M. Evans.

**Writing – original draft:** Luke M. Evans.

**Writing – review & editing:** Luke M. Evans, Christopher H. Arehart, Andrew D. Grotzinger, Travis J. Mize, Maizy S. Brasher, Jerry A. Stitzel, Marissa A. Ehringer, Charles A. Hoeffer.

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
