## [Decision Letter · Decision Letter 0]

1 Nov 2022

Dear Dr EVANS,

Thank you very much for submitting your Research Article entitled 'Transcriptome-wide gene-gene interaction associations elucidate pathways and functional enrichment of complex traits' to PLOS Genetics.

The manuscript was fully evaluated at the editorial level and by independent peer reviewers. The reviewers appreciated the attention to an important problem, but raised some substantial concerns about the current manuscript. Based on the reviews, we will not be able to accept this version of the manuscript, but we would be willing to review a much-revised version. We cannot, of course, promise publication at that time.

If you decide to revise the manuscript for further consideration at PLOS Genetics, please aim to resubmit within the next 60 days, unless it will take extra time to address the concerns of the reviewers, in which case we would appreciate an expected resubmission date by email to plosgenetics@plos.org.

We are sorry that we cannot be more positive about your manuscript at this stage. Please do not hesitate to contact us if you have any concerns or questions.

Yours sincerely,

Yun Li

Academic Editor

PLOS Genetics

Xiaofeng Zhu

Section Editor

PLOS Genetics

Reviewer's Responses to Questions

**Comments to the Authors:**

Reviewer #1: Enclosed is a review of Evans et al's manuscript

"Transcriptome-wide gene-gene interaction associations elucidate pathways and functional enrichment of complex traits."

In this manuscript, the authors develop a framework to measure epistasis by employing a flexible approach based on genetically-regulated expression and TWAS-based approaches. The method identifies and replicates some "hub" genes with multiple interactions. The method is ambitious and important and is a necessary step towards interrogating gene-gene interactions and their effects on complex traits. Overall, multiple questions still remain about the functionality of the method and the interpretation of the results.

1. Line 144 - A natural question here is whether or not we expect epistatic genes to be correlated, even on the genetically-imputed scale. Can the authors comment on this and provide some intuition? It does make sense that correlation between the two variables (T1 and T2) or shared SNPs underlying the predictive model can lead to some level of collider bias, but do we not want to pick these genes up?

2. There are multiple different thresholds for correlations and P-values used throughout the paper. I would prefer the authors to aggregate these in a single figure/table and justify further.

3. For replication of findings, only different GWAS were used. Since these cross-genome interactions may be affected by cell-type heterogeneity, replication across different weights may be important. Perhaps the authors could use the PsychENCODE weights and GTEx weights on the same GWAS to replicate?

4. Does the interaction term need to also be residualized by covariates? I suggest the authors run a sensitivity analysis for false positives/power by looking at residualizing the T1 x T2 term, as well, or provide some rationale otherwise.

5. Can the authors discuss an adaptive strategy to perhaps decrease the number of tests run?

6. Figure 2 is a little hard to understand. Do peaks in this Manhattan plot bear the same meaning as they do for GWAS/TWAS? For example, there's a peak on Chromosome 19 in Figure 2 (top). Do the authors consider this to be a hub gene?

7. A massive limitation of the paper is the lack of associated scripts. This is necessary for any level of replicability, especially when the scalability of the method is so important. Does the tool include ways to parallelize across other machines, or is this specific to the environments used by the authors?

Reviewer #2: This is a well written and straightforward paper. The authors propose a new way to detect gene-gene interactions (in a broad sense). Instead of looking for SNPs in genes that interact with each other to affect traits, they imputed gene expression from eQTLs and looked for expression of pairs of genes that interact to affect traits. The model was simple, an interaction term (product of gene expression) was tested for significance in the presence of main effects. This model was fitted for every pairs (minus those with high correlation) thus termed TWIS (transcriptome wide interaction study). Several interactions were discovered and replicated in multiple datasets. Furthermore, the authors developed a method to test for gene set enrichment (E-TWIS) and found many pathways and networks enriched for TWIS signals. These led the authors to conclude that epistasis is likely widespread and the proposed methods may offer a useful way to explore gene-gene interactions.

Overall I find the paper easy to read, though it's a bit dense in details. The question is definitely an important one, i.e. what is the contribution of gene interactions to complex traits. However, the model as specified in the paper is very limited in scope and finds only one particular type of interaction for genes that are expressed in one tissue at a time. While it may seem like a good alternative to the computationally intractable search for SNP-SNP interactions, the two test for completely different hypotheses. These should be clearly pointed out and the limitations discussed. I detail below a few major points that need to be addressed/discussed:

1) The SNP-SNP interaction tests look for DNA variants who interact to influence traits. The gene-gene expression interaction tests look for gene expression that interact to influence traits. A genuine SNP-SNP interaction is causal, a genuine gene-gene expression may not be causal and may reflect only secondary and reactive effects from the traits.

2) The model tests for significance of the variable T1*T2. Because the imputed gene expression is based on an additive eQTL model, this means the proposed TWIS approach only discovers additive x additive interactions. There are a lot more types of interactions than additive x additive that would be missed by TWIS.

3) Line 215: It is not appropriate to test for a main effect when the interaction term is in the model. In the presence of a significant interaction, the main effect is meaningless because it's context dependent (depending on the other gene). The main effect is only relevant when there is no other term that includes it in the model.

4) I suggest the authors add plots to visualize the association between the top interactions and traits. For example, a 3-D plot with the x, y representing the two genes and the z representing the trait may be warranted here. Alternatively, plot phenotype against T1*T2.

5）Figure 1. Both the traits and gene expression were adjusted for the same set of covariates. This may create spurious association. It's important to evaluate this in simulation. There are several scenarios: covariates have effects on only traits, only expression, no effects on either, effects on both. But there is not association between traits and expression, would you find false associations if both are adjusted by the same set of covariates.

6) Line 631: I'm not sure I agree that the sum of m Z scores in this context is a chi-squared with m df. The m Z scores are obviously not independent. This effect is more pronounced when the gene set is large and the authors propose a secondary resampling approach to guard against false positives. However, I think this should be the primary approach to be used for all gene sets, regardless of their size.

Reviewer #3: In this manuscript, Evans and coauthors introduce a new approach named TWIS to find gene-gene interactions affecting complex traits. This extends the TWAS approach by testing pairwise interaction effects between imputed gene expression on complex traits of interest. The main advantage of the proposed approach compared to existing methods is the reduced multiple testing burden (due to the test being at the gene-level rather than the variant-level) without making any pre-filtering based on, for example, the significance of a single gene analysis. The authors apply their proposed approach to 12 complex traits using large datasets and found a few significant interactions. The authors also developed a procedure to test for enrichment of gene-gene interactions in predefined sets. The results are somewhat underwhelming, given that only 1 interaction replicated in an independent dataset and only 6 interactions were significant at the meta-analysis across datasets. However, it is refreshing to see researchers focus on gene-gene interactions in humans, which have been documented extensively in model organisms, rather than neglecting them a priori. Furthermore, the authors performed extensive simulations to show that their approach controls for false positives, but has low power (as expected for interaction effects) at current sample size. I believe this is a promising approach and is of interest to the broad human genetics community, however, I do have a few comments.

1 – Line 65-66. I think it is important to distinguish between gene action and contribution to variance components. See Huang and Mackay 2016 PLOS Genetics

2 – Line 108. What was the rational for choosing these particular traits?

3 – Line 115-119. I am not sure about the importance of PFC for height and BMI, for example. So, the results for those traits may be affected negatively by the choice of these tissues. Maybe an enrichment-type analysis (e.g., S-LDSC) could be performed to choose the most relevant tissues?

4 – Line 134. Is it really N(0,1) or is it N(0, sigma^2)?

5 – Line 142. I am assuming you mean LD rather than linkage? I would also point out that LD is between the variants constituting the expression prediction model. However, it is important to note that LD is only one possible source of imputed expression correlation. From Weinberg et al 2019 Nat Gen “A gene pair can have correlated predicted expression if the same causal eQTL regulates both genes or if two causal eQTLs in LD each regulate one of the genes”. In general, I find that LD and linkage – two very distinct concepts – are sometimes used interchangeably throughout the manuscript adding to the confusion. Please make sure that the correct term is used in the appropriate context.

6 – Line 146-148. Are there cases where |r| < 0.05 within 1 Mb? And if so, could such cases lead to false positives? My understanding is that physical distance per se doesn’t inflate false positive rate — it is the high correlation that does inflate, which may be due to physical distance. So, the authors could remove the filter based on distance?

7 – Line 154-163. I find this part pretty confusing. For example, there are 4 significant interactions for pAUDIT in Table 1 but only 3 interactions above the dashed line in Fig 2. Also, in Fig. 2 either the authors present all the significant results (i.e., including SC too) or they pick only one trait as an example. Also, on line 162 the authors say “Of the five significant in the final meta-analysis…”, but they are 6, if I understand correctly. Please make the whole paragraph clearer.

8 – Line 186. I am not sure what you mean by “imperfect” expression imputation. If it is r^2 < 1, that will never happen for genes with h^2_g < 1. Remember that r^2 <= h^2_g since only SNPs are used to predict expression. So, you could have r^2=h^2_g<1 which would be perfect imputation. So, I would define imperfect imputation as r^2<h^2_g. clearly="" definition="" in="" please="" state="" text.="" the="" your="">

9 – Line 190. Fig. 3 says p<1e-6. Which is the correct one?

10 – Line 211-227. If I understand correctly, these are two different situations. The first (GRIK1 - CPD association) is an example of a main effect not being identified without an interaction in the model (probably because the interaction explained some variance that went into the error in the single gene model). The second (% of genes recovered by the single gene model) is an example of genes that have a significant interaction association without their main effects being significant. If my understanding is correct, this is comparing apples to oranges. A better comparison would be checking the % of genes with significant MAIN effect at TWIS that can be recovered by TWAS. If my understanding is incorrect, please clarify.

11 – Line 239-242. Why did you choose these specific gene sets?

12 – Line 276. Not sure you can conclude that from those two observations, especially given that the actual significant interactions are only 6 and for 3 of the 12 traits analyzed.

13 – Line 422. From the fastLm manual “However, Armadillo will either fail or, worse,

produce completely incorrect answers on rank-deficient model matrices whereas the functions from the stats package will handle them properly due to the modified Linpack code”. This might not have been a problem in your analysis, but is something to keep in mind when testing for interactions.

14 – Line 504-505. I think you mean decreasing interaction PVE also decreases the power. And adding prediction error further decreases power. Is that right?

15 – Line 523. Genes are not in LD — variants are. Again, please make sure the appropriate terms are used.

16 – Could the authors try to find the variant-variant interaction(s) underlying the significant gene-gene interactions? For example, by testing all the possible pairwise interactions between the variants making up the prediction models for the two genes. The significance threshold would be reduced like in a candidate gene approach. Power might still be an issue but it is worth trying.>/h^2_g.>

**Have all data underlying the figures and results presented in the manuscript been provided?**

Reviewer #1: Yes

Reviewer #2: Yes

Reviewer #3: Yes

PLOS authors have the option to publish the peer review history of their article (what does this mean?). If published, this will include your full peer review and any attached files.

Reviewer #1: No

Reviewer #2: No

Reviewer #3: No

---

## [Decision Letter · Decision Letter 1]

31 Jan 2023

Dear Dr Evans,

Thank you very much for submitting your Research Article entitled 'Transcriptome-wide gene-gene interaction associations elucidate pathways and functional enrichment of complex traits' to PLOS Genetics.

The manuscript was fully evaluated at the editorial level and by independent peer reviewers. The reviewers appreciated the attention to an important topic but identified some concerns that we ask you address in a revised manuscript.

We therefore ask you to modify the manuscript according to the review recommendations. Your revisions should address the specific points made by the reviewers in terms of visualization and correct usage of the LD term.

Yours sincerely,

Yun Li

Academic Editor

PLOS Genetics

Xiaofeng Zhu

Section Editor

PLOS Genetics

Reviewer's Responses to Questions

**Comments to the Authors:**

Reviewer #1: I commend the authors for going above and beyond addressing my comments.

Reviewer #2: The authors have adequately addressed most of my questions. However, when I asked (#4) for visualization of the significant interactions, I suggested 3D plot or 2D plot (phenotype versus T1*T2). I expected the authors to at least make some effort to make sense of the plot. They did make a 3D plot but did not try to explain any of it other than saying there is a figure. I believe this is a lost opportunity to make the paper more accessible. I suggest the author in their text explain how the figure is showing association between the phenotype and T1*T2. It does not have to be a 3-D plot, which apparently isn't very intuitive. Perhaps phenotype ~ T1*T2? or any other ways you find useufl to visualize the association.

Reviewer #3: I thank the authors for addressing my comments -- I am generally satisfied with the revision. One minor issue though. There are still a few places in the manuscript where the authors say "genes in LD", for example lines 584, 610, 611, 622, 663, 668. Variants are in LD, not genes. Please correct.

**Have all data underlying the figures and results presented in the manuscript been provided?**

Reviewer #1: Yes

Reviewer #2: Yes

Reviewer #3: Yes

PLOS authors have the option to publish the peer review history of their article (what does this mean?). If published, this will include your full peer review and any attached files.

Reviewer #1: No

Reviewer #2: No

Reviewer #3: No

---

## [Editor Report · Decision Letter 2]

6 Mar 2023

Dear Dr Evans,

We are pleased to inform you that your manuscript entitled "Transcriptome-wide gene-gene interaction associations elucidate pathways and functional enrichment of complex traits" has been editorially accepted for publication in PLOS Genetics. Congratulations!

Yours sincerely,

Xiaofeng Zhu

Section Editor

PLOS Genetics

Xiaofeng Zhu

Section Editor

PLOS Genetics

Comments from the reviewers (if applicable):

**Data Deposition**

http://datadryad.org/submit?journalID=pgenetics&manu=PGENETICS-D-22-01076R2

**Press Queries**

---

## [Editor Report · Acceptance letter]

16 May 2023

PGENETICS-D-22-01076R2 

Transcriptome-wide gene-gene interaction associations elucidate pathways and functional enrichment of complex traits 

Dear Dr Evans, 

We are pleased to inform you that your manuscript entitled "Transcriptome-wide gene-gene interaction associations elucidate pathways and functional enrichment of complex traits" has been formally accepted for publication in PLOS Genetics! Your manuscript is now with our production department and you will be notified of the publication date in due course.

With kind regards,

Marianna Bach

PLOS Genetics

On behalf of:
